# Strong bulk photovoltaic effect in engineered edge-embedded van der Waals structures

Zihan Liang[1,7], Xin Zhou[2,7], Le Zhang[1,7], Xiang-Long Yu[3,4] ✉, Yan Lv[5], Xuefen Song[5], Yongheng Zhou[1], Han Wang[1], Shuo Wang[5], Taihong Wang[1], Perry Ping Shum[1], Qian He[2], Yanjun Liu[1], Chao Zhu[6], Lin Wang[5] ✉ & Xiaolong Chen[1] ✉

Bulk photovoltaic effect (BPVE), a second-order nonlinear optical effect governed by the quantum geometric properties of materials, offers a promising approach to overcome the Shockley-Quiesser limit of traditional photovoltaic effect and further improve the efficiency of energy harvesting. Here, we propose an effective platform, the nano edges embedded in assembled van der Waals (vdW) homo- or hetero-structures with strong symmetry breaking, low dimensionality and abundant species, for BPVE investigations. The BPVE-induced photocurrents strongly depend on the orientation of edge-embedded structures and polarization of incident light. Reversed photocurrent polarity can be observed at left and right edge-embedded structures. Our work not only visualizes the unique optoelectronic effect in vdW nano edges, but also provides an effective strategy for achieving BPVE in engineered vdW structures.

Efficient conversion of light to electricity lies at the heart of eco-friendly energy harvesting. Extrinsic photovoltaic effect, utilizing the built-in electric field in p-n junctions or photo-thermoelectric effect, has been widely studied for many decades and its efficiency starts to reach the theoretical limit[1,2]. Alternatively, bulk photovoltaic effect (BPVE) or intrinsic photogalvanic effect, a second-order nonlinear optical effect arising from the broken inversion symmetry of materials, can overcome this efficiency barrier, since it does not rely on the extrinsic electric field or thermal gradient[3–13]. Initial studies on BPVE have been mainly focused on ferroelectric insulators with broken inversion symmetry[14–20]. Recent studies suggest that Weyl semimetals and narrow-bandgap semiconductors, such as TaAs and tellurium, can support stronger BPVE[21–25]. Other strategies, such as lowering lattice

symmetry and dimensionality, were also proposed to effectively enhance BPVE[3–6,12,26–35]. For example, BPVE-induced photocurrent density in one-dimensional (1D) $WS_2$ nanotube is over $1\,A\,cm^{-2}$, which is among the highest reported values[3]. Van der Waals (vdW) layered materials, with low dimensionality, rich species, and good flexibility, offer another ideal platform to investigate BPVE. Although majority vdW materials possess lattice inversion symmetry, several approaches have been developed to generate strong BPVE. For example, strain gradient in $2H-MoS_2$ and an uniform strain in rhombohedrally stacked $MoS_2$ ($3R-MoS_2$) can break the lattice inversion symmetry and induce giant in-plane short-circuit photocurrents under linearly polarized light[4,36]; Moiré-pattern-induced symmetry breaking in twisted bilayer graphene allow the observation of pronounced BPVE[5]; Interface of

[1]Department of Electrical and Electronic Engineering, Southern University of Science and Technology, Shenzhen, China. [2]Department of Materials Science and Engineering, National University of Singapore, Singapore, Singapore. [3]Shenzhen Institute for Quantum Science and Engineering, Southern University of Science and Technology, Shenzhen, China. [4]International Quantum Academy, Shenzhen, China. [5]School of Flexible Electronics (Future Technologies) & Institute of Advanced Materials (IAM), Key Laboratory of Flexible Electronics (KLOFE), Jiangsu National Synergetic Innovation Center for Advanced Materials (SICAM), Nanjing Tech University (NanjingTech), Nanjing, China. [6]SEU-FEI Nano-Pico Center, Key Laboratory of MEMS of Ministry of Education, Collaborative Innovation Center for Micro/Nano Fabrication, Device and System, Southeast University, Nanjing, China. [7]These authors contributed equally: Zihan Liang, Xin Zhou, Le Zhang. ✉e-mail: yuxl@sustech.edu.cn; iamlwang@njtech.edu.cn; chenxl@sustech.edu.cn

black-phosphorus/WSe$_2$ heterostructure generates in-plane strong polarization and directional photocurrents in WSe$_2$ due to the broken inversion symmetry[12]; The spontaneous polarization in out-of-plane direction was also observed in 3R-MoS$_2$[37]; Non-centrosymmetric nano-antennas can assist the generation of artificial BPVE in centrosymmetric graphene flakes[6,33,34,38]; Berry curvature dipole in monolayer topological insulator WTe$_2$ also support BPVE and can be controlled with a vertical displacement field[39]. On the other hand, nano edges, where the periodic crystalline structure of a material is interrupted, show broken inversion symmetry and theoretically can host BPVE[21,40,41]. Recently, BPVE-induced photocurrents traveling along specific edges were experimentally reported in type-II Weyl semimetal WTe$_2$[21]. The edge photocurrents strongly depend on the geometric symmetry near edges and are possibly enhanced by fermi-arc type surface states[21]. These results suggest vdW nano edges a promising platform for BPVE investigations. However, for majority vdW materials, including MoS$_2$ and ReS$_2$, BPVE in nano edges is either negligible or indistinguishable from extrinsic effects.

Here, we discover strong BPVE in engineered low-symmetric 1D vdW nano edges through constructing edge-embedded homo- or hetero-structures. The edge-embedded structure shows lower geometric symmetry and distinct local properties, which could support a strong BPVE. Our strategy is unique compared with previous demonstrations in two aspects. Firstly, the proposed edge-embedded vdW structures potentially have thousands of configurations, since they can be conveniently constructed using various vdW layered materials. The abundant species of edge-embedded vdW structures provide a platform for physics and device engineering. Secondly, symmetry analysis suggest that the BPVE-induced photocurrents should have reversed

polarity at left and right edge-embedded structures, while extrinsic-photovoltaic-effect-induced photocurrents have the same polarity. Thus, this geometric symmetry of left and right edges embedded in one photodetection device, allows us to approximately identify the BPVE-induced photocurrent from the extrinsic one.

## Results

### BPVE in ReS$_2$/ReS$_2$ edge-embedded structures

VdW materials, including ReS$_2$, MoS$_2$, and WS$_2$, are mechanically exfoliated from bulk crystals using the "scotch tape" method. Then, the edge-embedded vdW structures are assembled on SiO$_2$/Si substrates using the PDMS-assisted dry-transfer method[42]. The edge-embedded structure is regarded to the nearby region of bottom edge (see Fig. 1a, b) and top flake as schematically shown in Fig. 1c, d. The physical properties of edge-embedded structure should be different from those of bulk, since quasi-1D edge states and strains exist in this local region. From this point of view, we can treat the edge-embedded structure as quasi-1D. Figure 2a shows the optical image of a fabricated ReS$_2$/ReS$_2$ vdW homostructure device. At the edges of bottom flakes, roof structures of top flakes are self-assembled as a result of the balance of flexibility and stiffness of vdW materials. In following sections, we will show that the edge-embedded structures exhibit 1D confined optoelectronic properties, similar to those of 1D WS$_2$ nanotubes[3]. The cross-sectional scanning transmission electron microscope (STEM) is further performed to investigate edges and edge-embedded structures. As shown in Fig. 2c–f, surfaces of vdW edges of mechanically exfoliated ReS$_2$ flakes are not perpendicular to the substrate surface. Instead, inclined angle, cracks and dangling structures are observed in vdW edges. The broken inversion symmetry of vdW edges is supposed

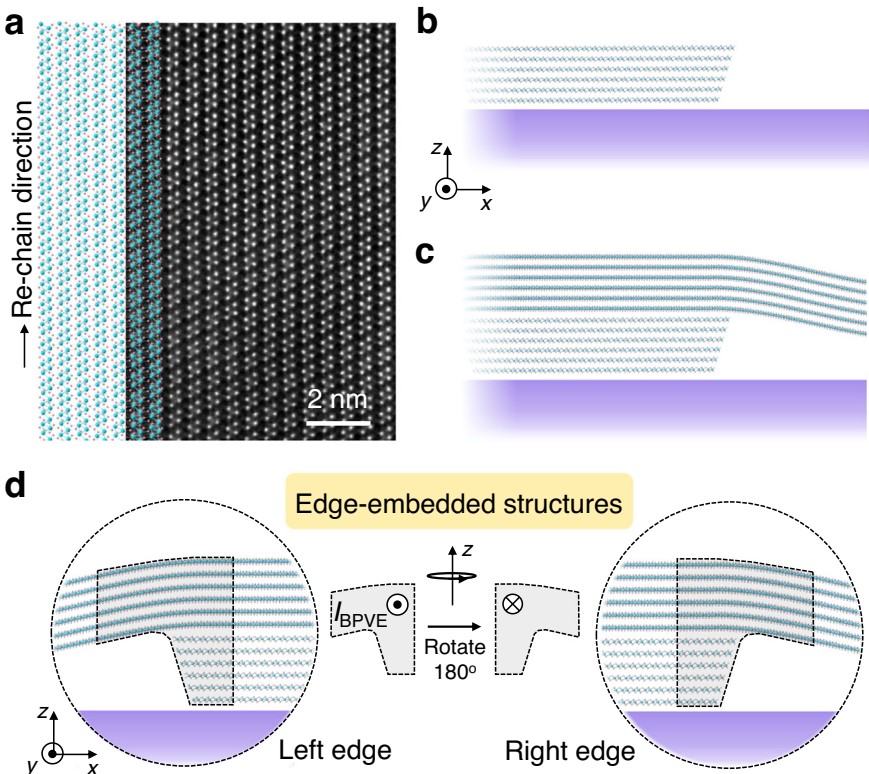

**Fig. 1 | Schematics of van der Waals (vdW) edge-embedded structures.**
**a** Scanning transmission electron microscope image (STEM) and schematic of ReS$_2$ crystal from top view. **b** Cross-sectional schematic of nano edges in mechanically exfoliated vdW layered materials. Purple region represents the substrate.
**c**, **d** Cross-sectional schematic of edge-embedded vdW homo- or hetero-structures proposed in this work. Effective vdW coupling between bottom edge regions and

top vdW materials can result in strong symmetry-breaking and quasi-one-dimensional edge-embedded structures, which host strong bulk photovoltaic effect (BPVE). Symbols "⊙" and "⊗" represents the $y$ and $-y$ direction of BPVE-induced photocurrents $I_{BPVE}$, respectively. The dashed-line enclosed area schematically shows the whole edge-embedded structure.

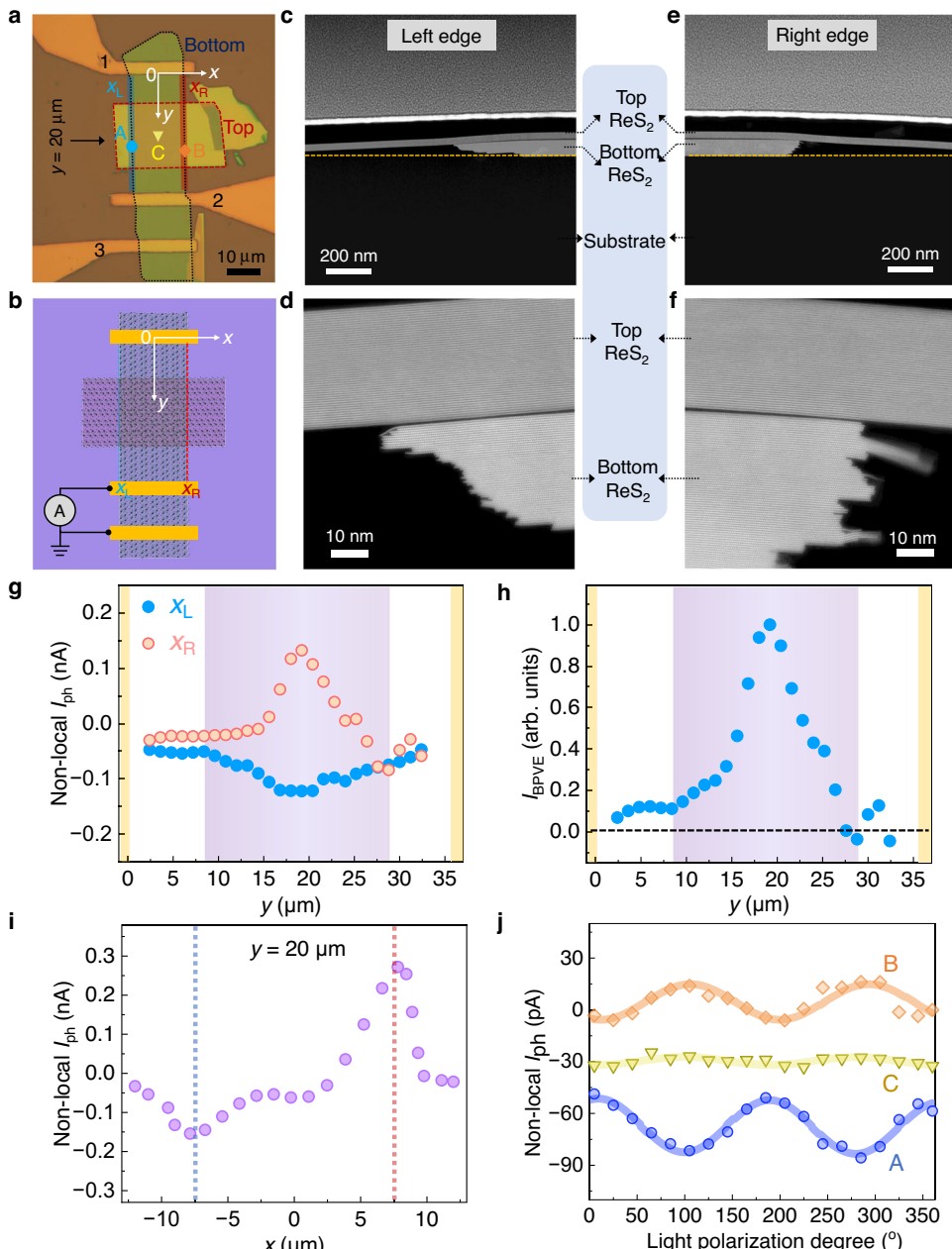

**Fig. 2 | Characterization of a ReS$_2$/ReS$_2$ homostructure device. a** Optical image of the ReS$_2$/ReS$_2$ homostructure device. Left ($x_L$) and right ($x_R$) edges, including edge-embedded structures, are denoted by blue and red solid lines. The bottom and top ReS$_2$ flakes are enclosed by black and red dashed lines, respectively. Scale bar is 10 μm. **b** Schematic of the short-circuit non-local photocurrent measurements. **c–f** Cross-sectional STEM of edge-embedded structures in the ReS$_2$/ReS$_2$ homostructure. **d** and **f** show the enlarged regions of left and right edge-embedded structures, respectively. **g** Non-local photocurrents, detected by electrodes 2 and 3, along left (blue dots) and right (orange dots) edges. Purple shaded region shows the position of edge-embedded structures. Yellow shaded regions show the

position of electrodes. Opposite-directional photocurrents are observed along left and right edge-embedded regions. **h** Normalized BPVE-induced photocurrent along edges using $I_{BPVE} = \pm (I_{ph}(x_L) - I_{ph}(x_R))/2$. A peak feature is observed in the edge-embedded regions. **i** Non-local photocurrents along $x$ direction at $y = 20$ μm. Blue and red dashed lines show the position of vdW edges. **j** Linear-polarization-dependent non-local photocurrents at three representative positions A, B, and C, marked in **a**. Dots are experimental results and solid lines are fitting curves using the expression $A\sin(2\theta + \phi)$, where $A$ is the amplitude and $\phi$ is the phase. Light polarization degree $\theta$ is the angle between light polarization and $x$-direction.

---

to generate BPVE current without any external voltage bias under polarized light (see symmetric analysis in Supplementary Note 1 and Supplementary Figs. 1 and 2). However, BPVE photocurrents at pure vdW edges, such as ReS$_2$ and MoS$_2$, are either negligible or mixed up with extrinsic photovoltaic effects (see data below). On the other hand, our proposed edge-embedded vdW structures further couple vdW edges with vdW materials, resulting in "⌐" or "⌐" shaped structures (see schematics in Fig. 1d and STEM results in Fig. 2c–f). This further

lowers the geometric symmetry of vdW edges and enhances the BPVE photocurrents.

Short-circuit non-local photocurrent measurements, where edge-embedded structures locate outside the channel, are mainly used to study the BPVE (see Fig. 2b), since non-local photocurrent improves the resolution of BPVE-induced photocurrents compared with local photocurrent (please see Supplementary Figs. 3–5 and Supplementary Note 2 for details about local photocurrent results

in this sample. Similar BPVE effect can be observed using local photocurrent measurements). For the ReS$_2$/ReS$_2$ homostructure device (see Fig. 2a), the thickness of bottom and top ReS$_2$ flakes are 79 and 32 nm, respectively, determined by STEM. The linear current-voltage characteristics indicate a good Ohmic contact between electrodes and ReS$_2$ (see Supplementary Fig. 4). Then, a 532 nm laser with radius ~2.5 μm is focused on the sample. The laser polarization direction is set to y-direction. Non-local photocurrents along $x_L$ and $x_R$ edges (denoted by blue and red dashed lines in Fig. 2a) in short-circuit configurations are probed by electrodes 2 and 3, and electrode 1 is floating. As shown in Fig. 2g, opposite-directional photocurrents are observed in left and right edge-embedded structures (indicated by the purple shaded area). Position-dependent photocurrents ($I_{ph}$ ~ y) at one edge bend upwards while photocurrents at the other edge bend downwards. If we assume the device geometry is perfectly symmetric, the contribution of extrinsic photocurrents at $x_L$ and $x_R$ edges should be the same. Then averaged BPVE-induced photocurrents can be approximately obtained using $I_{BPVE} \approx \pm (I_{ph}(x_L) - I_{ph}(x_R))/2$, which excludes the extrinsic contribution. Because extrinsic contribution $I_{Extrinsic}(x_L)$ and $I_{Extrinsic}(x_R)$ at left and right edges are unknown, we are unable to get the BPVE-induced photocurrents unless making assumption that left and right edges are symmetric with $I_{Extrinsic}(x_L) = I_{Extrinsic}(x_R)$ and $I_{BPVE}(x_L) = -I_{BPVE}(x_R)$. In our experimental design, we try best to minimize the asymmetric effect through choosing samples with parallel edges and fabricating electrodes perpendicular to edges. The estimated BPVE-induced photocurrent $I_{BPVE} \approx \pm (I_{ph}(x_L) - I_{ph}(x_R))/2$ should be at the same order with the real value. As shown in Fig. 2h, a peak feature of $I_{BPVE}$ appears in the edges embedded in vdW structures, while vanishing $I_{BPVE}$ are observed in pure edges outside the edge-embedded vdW structures. This demonstrates that the edge-embedded structures host strong BPVE. This peak feature is similar to that observed in 1D WS$_2$ nanotubes[3]. When laser is focused near the boundary of pure-edge/edge-embedded-structure, edge-embedded structures absorb less photons leading to the reduction of photocurrents away from the center. This is the possible reason that $I_{BPVE}$ curve has a peak feature. For peak positions, they vary from sample to sample since they depend on many factors including characterization methods and device geometry. Then, we map the photocurrent at y = 20 μm along x-direction (see Fig. 2i). Photocurrent valley and peak are observed at left (x = −7.5 μm) and right (x = +7.5 μm) edges embedded in vdW structures, respectively. Photocurrent mapping result is further shown in Supplementary Fig. 6. Although asymmetry at two edge-embedded structures may lead to discrepancy of photocurrent shapes ($I_{ph}$ ~ y), the trend should be similar (see Supplementary Fig. 7). The peak/valley features cannot be attributed to extrinsic photocurrents arising from photo-thermoelectric effect or built-in electric fields in Schottky junctions[3–6,36]. On the other hand, the existence of a 180° rotation symmetry along z-axis between left and right edge-embedded structures (see Fig. 1d) results in reversed photocurrent polarities at two edges agreeing well with the BPVE picture.

Symmetry analysis suggest that pure ReS$_2$ edges can support BPVE-induced photocurrent along edge directions. Details are shown in Supplementary Note 1 and Supplementary Figs. 1 and 2. However, we did not observe detectable signal along pure ReS$_2$ edges experimentally, probably due to the small value of second-order nonlinear DC photoconductivity tensor $\chi_{ijk}$ or narrow edge regions influenced by the broken inversion symmetry. Here, $i/j/k$ represents $x$, $y$, or $z$. On the other hand, although inversion symmetry breaks near the ReS$_2$ edge, the inversion symmetry is still approximately preserved in unit cell at edges which could probably significantly reduce the BPVE signal. For ReS$_2$ edge-embedded structures, the local geometric symmetry is further lowered which allows a y-directional photocurrent density

$J_y^{LBPVE} = D + C \cos(2\alpha + \varphi)$, where $\alpha$ is the angle between x-direction and light polarization direction, $D = (\chi'_{yxx} + \chi'_{yyy})E_0^2$ is the polarization-independent term, $C = \sqrt{(\chi'_{yxx} - \chi'_{yyy})^2 + 4\chi'_{yxy}{}^2}E_0^2$, and $\varphi$ is the phase. However, the second-order nonlinear DC tensor $\chi'_{ijk}$ of edge-embedded structures should be different from that of pure edges $\chi_{ijk}$, since it depends not only on the type of materials but also on the geometry. For example, monolayer WS$_2$ flakes and WS$_2$ nanotubes are both non-centrosymmetric materials which theoretically support BPVE[3]. However, pronounced BPVE-induced photocurrents are only observed in WS$_2$ nanotubes[3]. This suggests that the second-order nonlinear DC tensor of WS$_2$ nanotube should be significantly larger than that of monolayer WS$_2$ flake. Hence, it is reasonable that $\chi'_{ijk}$ is much larger than $\chi_{ijk}$, since the geometry of engineered edge-embedded structures is different with that of pure edges.

Strong polarization-dependence is another key feature of BPVE-induced photocurrents[3–6,36]. At homostructure region (position C), the photocurrent is almost independent of light polarization. The polarization ratio at position C is relatively weak which is probably due to the crystalline orientation of top and bottom ReS$_2$ flakes. For the ReS$_2$/ReS$_2$ homostructure sample, the Re-chain directions of bottom and top ReS$_2$ flakes are along y- and x- directions, respectively, demonstrated by cross-sectional STEM and Raman measurements (see Supplementary Fig. 8). The 90° difference of crystalline orientation and charge transfers between top and bottom flakes might decrease the polarization ratio at homostructure region. However, at edge-embedded vdW structure (position A and B), absolute values of photocurrents reach the maxima along the direction of vdW edges (along y-direction) with a large polarization ratio, showing distinct behavior compared with that at position C. This polarization phenomenon agrees well with that observed in 1D WS$_2$ nanotubes[3]. This indicates that the x-directional dimension of edge structures is confined, even they are embedded in vdW materials. Besides, we notice that the polarity of photocurrent varies with light polarization at position B. It is reasonable as the total photocurrent density can be expressed as $J_{total} = J_y^{LBPVE} + J_y^{Extrinsic} = D + C \cos(2\alpha + \varphi) + J_y^{Extrinsic}$. When $D + J_y^{Extrinsic} - C < 0$ is satisfied, the polarity of total photocurrent can vary with light polarization. This polarity switching phenomenon was also observed in previous BPVE studies in twisted bilayer graphene and Weyl semimetals[5,23]. We also notice that ReS$_2$ is an anisotropic semiconductor with maximum optical absorption along the Re-chain direction. In this sample, the ReS$_2$ edge is along y-direction which is coincidently the same with its Re-chain direction. In Fig. 3, we show another ReS$_2$/ReS$_2$ sample with different orientations of pure edge and edge-embedded structures. The Re-chain directions of bottom and top ReS$_2$ flakes are marked in Fig. 3. As shown in Fig. 3, bulk position (position D) and pure edge (position E) shows similar polarization response with maximum photocurrent near the Re-chain direction ($\theta$ ~ 30°), while for edge-embedded structures (position F), the maximum photocurrent response is along the y-direction/edge-direction ($\theta$ ~ 90°). This demonstrates that BPVE does influence the polarization properties at edge-embedded structures.

## BPVE in other vdW edge-embedded structures

To verify the universality of the concept of edge-embedded vdW structures and the observed BPVE effect, we repeat experiments in other edge-embedded homo- and hetero-structures, built from another two popular vdW materials MoS$_2$ and WS$_2$. Two representative structures, MoS$_2$/MoS$_2$ homostructure and WS$_2$/ReS$_2$ heterostructure (where WS$_2$ is on top), are fabricated and investigated (see Fig. 4a, e). Similar BPVE phenomena are observed in these devices. As shown in Fig. 4b, f, non-local photocurrents along two edges show peak/valley features in edge-embedded vdW structures (purple shaded area). $I_{BPVE}$ shows peak features in edge-embedded

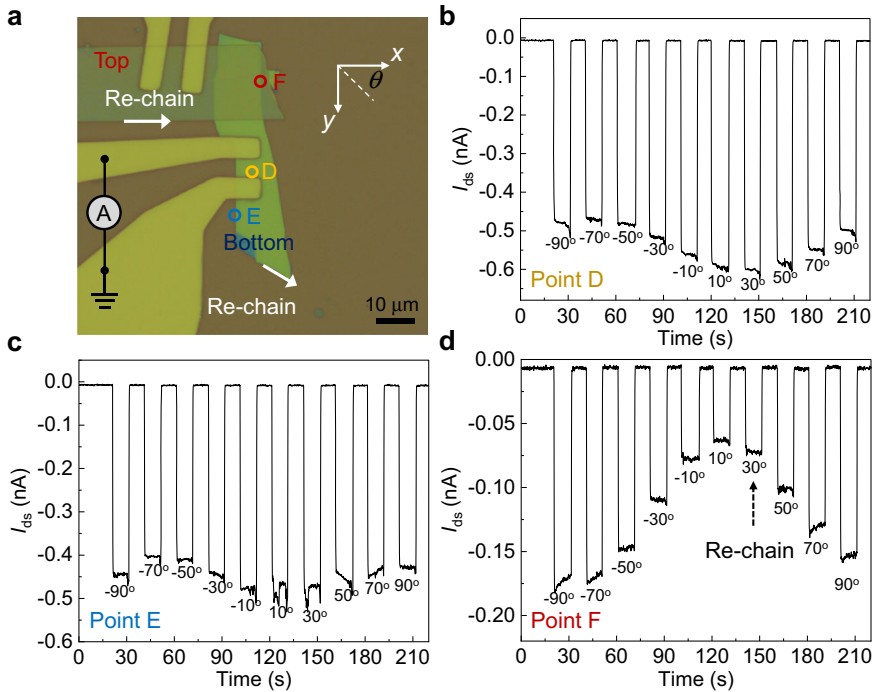

**Fig. 3 | Linear-polarization-dependent photocurrents in another ReS$_2$/ReS$_2$ sample. a** Optical image of another ReS$_2$/ReS$_2$ homostructure device. Scale bar is 10 μm. **b**–**d** Total measured current at bulk region (position D) **b**, pure edge (position E) **c**, and edge-embedded structures (position F) **d**. Laser is switched on and off every 10 s with different polarization angle $\theta$.

vdW structures and gradually vanishes in pure edges (see Fig. 4c). The larger extrinsic photocurrents near electrodes in MoS$_2$/MoS$_2$ device are probably due to the larger Schottky barrier in metal contact regions (see Supplementary Fig. 9). Supplementary Fig. 10 further shows the scanning photocurrent spectroscopy of the WS$_2$/ReS$_2$ device. We notice that signs of photocurrents at $x_L$ and $x_R$ are different for ReS$_2$/ReS$_2$ and WS$_2$/ReS$_2$ edge-embedded structures. We infer that the second-order nonlinear DC tensor $\chi'_{ijk}$ plays an important role in determining the directions of $I_{BPVE}$. As discussed above, $\chi'_{ijk}$ strongly depends on the geometry and orientations of edge-embedded structures. For the WS$_2$/ReS$_2$ edge-embedded structure, the configuration is different with that of ReS$_2$/ReS$_2$. Hence, it is possible that the sign and value of $\chi'_{ijk}$ in WS$_2$/ReS$_2$ edge-embedded structure is different from that in ReS$_2$/ReS$_2$ edge-embedded structures, resulting in different photocurrent signs. We further investigate polarization-dependent photocurrents in the MoS$_2$/MoS$_2$ homostructure device. As shown in Fig. 4d, no polarization dependence is observed outside edge-embedded region which is consistent with the isotropic crystalline structure of MoS$_2$. However, at edge-embedded region, polarization dependence is observed and the maximum value is along the edge direction. This provides another evidence that BPVE exists at edge-embedded structures.

## Discussion

At last, we discuss other possible scenarios that could generate the zero-bias photocurrents at edge-embedded structures. One argument is strain-gradient-induced photocurrents in top vdW flakes[4]. We agree that there exists strain gradient in top flakes along $x$-direction when top flakes step across the edge of bottom flakes with finite thickness. The strain gradient can break the centro-symmetry of materials and thus leads to polarization-dependent photocurrents. However, our electrodes are well defined and photocurrent collection direction is along $y$-direction, which is perpendicular to the direction of strain gradient. According to previous theoretical analysis, strain-gradient induced photocurrents along $y$-direction follow $\sin(2\theta)$ dependence,

where $\theta$ is the angle between strain gradient and light polarization direction[4]. Hence, strain-gradient induced photocurrents should vanish along $y$-direction ($\theta = 90°$), in sharp contrast to our observations in edge-embedded structures where photocurrents reach the maxima along $y$-direction. We further fabricate the ReS$_2$/hBN structure (see Fig. 5a), where the bottom hBN flake is an insulator, to investigate the contribution of $x$-directional strain gradient in ReS$_2$ to $y$-directional photocurrents. As shown in Fig. 5b, non-local photocurrents along two edges show similar trend without any peak/valley features, indicating a negligible contribution from the $x$-directional strain gradient. Similar results are observed in local photocurrent measurement (see Supplementary Fig. 11). To further illuminate the origin of BPVE in edge-embedded vdW structures, we fabricate the hBN/ReS$_2$ structure (see Fig. 5c), where hBN flake is on top. Similar $I_{ph}$ along two edges of hBN/ReS$_2$ structure (see Fig. 5d) indicates that coupling between bottom ReS$_2$ edges and top insulating material is not able to generate detectable BPVE. The edge region in hBN/ReS$_2$ structure is merely a pure edge with a different dielectric environment. Coupling between vdW semiconducting materials, where effective light-matter interactions can happen in the whole edge-embedded region, is also crucial to generate a strong BPVE photocurrent along edges as illustrated in Fig. 5e. We further did the first-principles calculations on ReS$_2$ edges and bent ReS$_2$ region. Pronounced edge states appear inside ReS$_2$ bandgap, showing different energy band properties from bulk and bent regions[21,40,41] (see Fig. 5f–h). Band bending near the interface of bottom edge and top bent region, even in homostructures, is able to generate effective charge transfer inside edge-embedded structures. Photo-excited charges tend to accumulate in the bottom edge due to the lower conductance band energy (see Fig. 5e). This is also the reason why we fabricate electrodes on bottom flakes. Besides, the charge transfers in $x$- and $z$- directions do not affect the $y$-directional momentum of carriers arising from BPVE in edge-embedded regions. Supplementary Fig. 12 further shows the calculated density of states in WS$_2$/ReS$_2$ edge-embedded structures, indicating a similar charge transfer process compared with that in ReS$_2$/ReS$_2$ edge-embedded

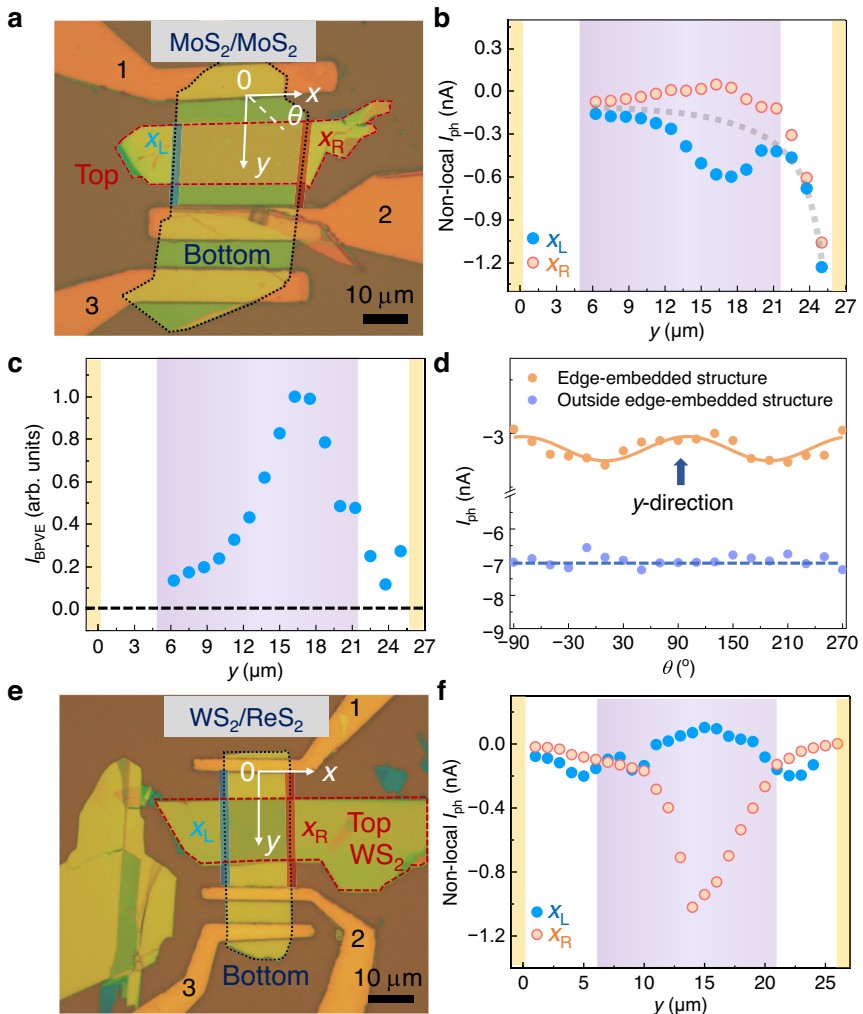

**Fig. 4 | Characterization of other homostructure and heterostructure devices.**
**a** Optical image of $MoS_2/MoS_2$ homostructure. Bottom and top $MoS_2$ flakes are enclosed by black and red dashed lines, respectively. **b**, **c** Non-local photocurrents along left ($x_L$) and right ($x_R$) edges **b**, and normalized $I_{BPVE}$ **c** in the $MoS_2/MoS_2$ homostructure device. Purple shaded region shows the position of edge-embedded vdW structures. Yellow shaded regions show the position of electrodes. Dashed gray line in **b** schematically shows the contribution of extrinsic photo-currents. Scale bar is 10 μm. **d** Polarization-dependent photocurrents at and out-side edge-embedded structure in the $MoS_2/MoS_2$ device. $\theta$ is the angle between

light polarization and $x$-direction. Solid line is the fitting curve using the expression $A\sin(2\theta + \phi)$, where $A$ is the amplitude and $\phi$ is the phase. Dashed line is the linear fitting curve of photocurrent outside the edge-embedded structure. **e** Optical image of $WS_2/ReS_2$ heterojunction. Bottom ($ReS_2$) and top ($WS_2$) flakes are enclosed by black and red dashed lines, respectively. **f** Non-local photocurrents along left ($x_L$) and right ($x_R$) edges in $WS_2/ReS_2$ heterostructure device. Scale bar is 10 μm. Purple shaded region shows the position of edge-embedded structures. Yellow shaded regions show positions of electrodes.

structures. It is worth noting that charge transfer processes are different for different edge-embedded structures. For other edge-embedded structures with different materials, charge transfer processes should be treated separately. Finally, Finite-Difference Time-Domain (FDTD) simulations are performed to compare the light absorption and power distribution in pure edges and edge-embedded structures (see Supplementary Fig. 13). No significant enhancement of light absorption is observed in edge-embedded regions.

In conclusion, through vdW coupling between bottom edge and top flake, we engineered an edge-embedded structure which has low geometric symmetry and distinct local properties. Under polarized light, photo-excited carriers acquire $y$-directional momentum governed by the BPVE in edge-embedded structures. Inside edge-embedded structures, photo-excited carriers tend to accumulated in bottom edges through charge transfer process. As a result, we are able to observe a detectable BPVE photocurrent along $y$-direction in edge-embedded structures. Other effects, such as photon drag effect, may also exist in low-dimensional structures, which is difficult to separate it

from BPVE[3,28]. On the other hand, edge-embedded vdW structures meet all the merits for observation of strong BPVE, including semi-conducting nature, low dimensionality and strong symmetry breaking. The observations are reproducible in various homo- and hetero-structures. Power-dependent photocurrents at edge-embedded vdW structures also show a transition from linear to square-root dependence, which agrees well with the "shift current" model for BPVE[3,4,36,43,44] (see Fig. 6). Hence, we conclude that edge-embedded vdW structures can host BPVE and offer a unique and versatile plat-form for investigating optoelectronic effects.

## Methods
### Sample preparation
The vdW homo- and hetero-structures were fabricated using the PDMS-assisted dry transfer method. Bottom and top flakes were mechanically exfoliated onto the $SiO_2/Si$ and PDMS/glass substrates, respectively. Then the top flake was aligned and transferred onto the bottom flake under a microscope. The electrodes were patterned

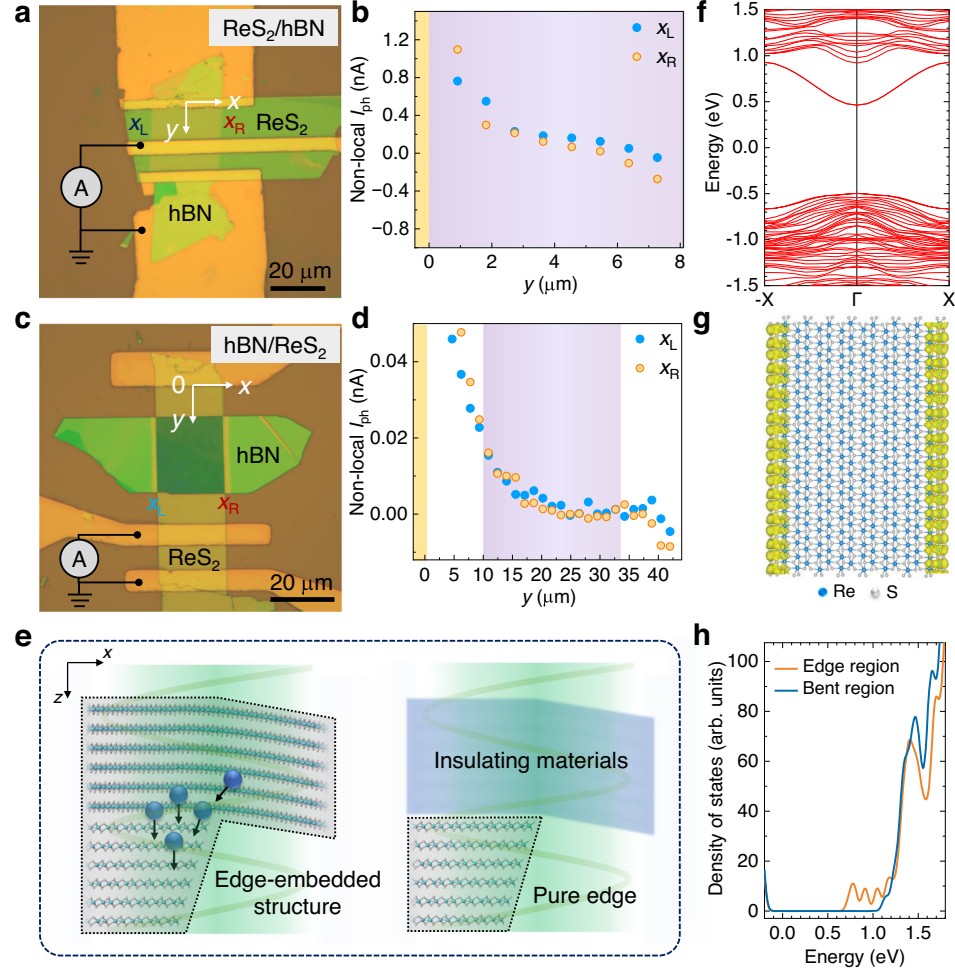

**Fig. 5 | Characterizations of hBN/ReS₂ and ReS₂/hBN heterostructure devices.**
**a, b** Optical image **a** and photocurrents $I_{ph}$ along edges **b** in the ReS₂/hBN structure. No valley and peak features along edges indicate that $x$-directional strain gradient does not contribute to BPVE along $y$-direction. Purple shaded region shows the position of edge-embedded vdW structures. Yellow shaded regions show the position of electrodes. Scale bar is 20 μm. **c, d** Optical image **c** and photocurrents $I_{ph}$ along edges **d** in the hBN/ReS₂ structure. Scale bar is 20 μm. **e** Left schematic shows that inside edge-embedded structures photo-excited charges with $y$-directional momentum tends to accumulate at bottom edges. Right schematic shows

that the top insulating material is not strongly coupled with the bottom edge, since it absorbs no light. The whole structure is merely a pure edge with different dielectric environment. **f** Calculated band structure of a ReS₂ nanoribbon with edge direction along the Re-chain direction. A new band from edges states appear inside the bandgap of bulk ReS₂. **g** Calculated edge states at the lowest conductance band edge (Γ point) are shown by the yellow region. The yellow color only shows the spatial distribution of edge states, which does not contain any magnitude information. Blue and gray dots represent Re and S atoms, respectively. **h** Calculated density of states of ReS₂ nano ribbon and bent ReS₂.

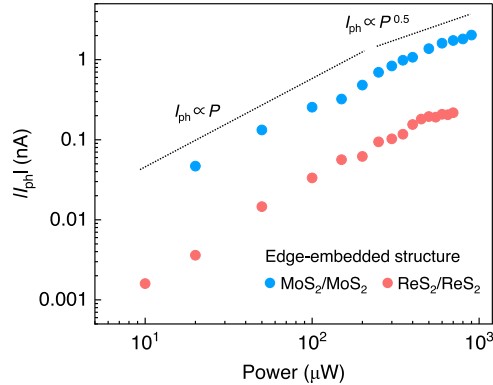

**Fig. 6 | Power-dependent photocurrent in edge-embedded structures.** Power-dependent photocurrents show a transition from linear to square-root dependence in edge-embedded structures in ReS₂/ReS₂ and MoS₂/MoS₂, which agrees well with the "shift current" model for BPVE. Dashed lines serve as guidelines for linear and square-root dependence.

using photolithography (Micro Writer ML3), followed by thermal deposition of Cr (3 nm)/Au (50 nm).

## Cross-sectional STEM characterizations

The cross-sectional lamellas of the devices were prepared by a focused ion beam system (FEI Helios Nanolab 450 S). To protect the sample and minimize the damage during ion beam milling, one layer of graphite was transferred to cover the interested device area, followed by a ~30 nm thick sputter coating platinum layer. Atomic-resolution STEM images were taken with a high-angle annular dark field (HAADF) detector at 200 kV, using JEOL ARM200F with CEOS aberration corrector. To align with the device configuration, the cross-sectional lamella direction was carefully controlled and marked during FIB milling and STEM observation. Alpha and beta angles of the well-tilted sample were spontaneously recorded when taking STEM images, to estimate the relative crystal orientation.

## Optical characterizations

For short-circuit local and non-local photocurrent measurements, devices were characterized using a semiconductor parameter analyzer

(FS-Pro) under vacuum (~$10^{-4}$ mbar) at room temperature. A continuous wave 532 nm laser with radius ~2.5 μm was used to excite the photocurrent. For polarization-dependent characterization, a half-wave plate was used to adjust the linear polarization of the incident laser. Raman spectra of the $ReS_2$, $MoS_2$ and $WS_2$ flakes were excited by a 532 nm laser and further collected by a Raman system (Zolix Finder Smart FST2-MPL501-405C1) with a spectrometer (Andor SR-500i-D2).

## First-principles calculations

The electronic structures were carried out using the local density approximation of the density-functional theory as implemented in the Vienna ab initio simulation package (VASP)[45]. The exchange-correlation potentials were described through the Perdew–Burke–Ernzerhof (PBE) functional within the generalized gradient approximation (GGA) formalism[46]. The cutoff energy for the expansion of the electronic wave functions was set at 400 eV and the tolerance of total energy is smaller than $10^{-4}$ eV. To qualitatively analysis physical mechanism, the monolayer was calculated for different flake configurations.

## Data availability

Relevant data supporting the key findings of this study are available within the article and the Supplementary Information file. All raw data generated during the current study are available from the corresponding authors upon request.

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

## Acknowledgements

The work was financially supported by the National Natural Science Foundation of China (62275117, X.C.; 92064010, L.W.), Shenzhen Excellent Youth Program (RCYX20221008092900001, X.C.), Shenzhen Basic Research Program (20220815162316001, X.C.), the National Key R&D Program of China (2022YFB3602801, L.W.; 2020YFA0308900, L.W.), Guangdong Major Talent Project (2019QN01C177, X.C.; 2019CX01×014, T.W.), the funding for "Distinguished professors" and "High-level talents in six industries" of Jiangsu Province (XYDXX-021, L.W.).

## Author contributions

X.C. and L.W. conceived and supervised the projects. Z.L. and L.Z. fabricated vdW homo- and hetero-structure devices with assistance of T.W. and P.S. Z.L. characterized photocurrent of devices with assistance of L.Z., Y.Z., H.W., Y.Lv, X.S., and S.W. X.Z., Q.H., and C.Z. performed the cross-sectional STEM characterizations and analysis. X.Y. performed the first-principles calculations and symmetry analysis. Z.L. and Y.L. did the Lumerical FDTD modeling. X.C. proposed the BPVE mechanisms. All authors discussed and commented the manuscript.

## Competing interests

The authors declare no competing interests.
