## [Peer Review File · Nature Communications]

Strong bulk photovoltaic effect in engineered edge-embedded van der Waals structuresREVIEWER COMMENTS

Reviewer #1 (Remarks to the Author):

In the manuscript, Liang et al. reported the experimental observation of non-local photocurrent response at the edge of ReS₂/ReS₂ homo-structure as well as several other vdWs structures. They attribute this to a strong bulk photovoltaic effect (BPVE) in the edge-embedded vdWs structures. BPVE has drawn lots of research interests in the past few years in order to find new candidates for optoelectronics applications. The experimental results are interesting. However, I think the experimental data cannot support that the photocurrent at the nano edge embedded in assembled VdW structure comes from BPVE. So I don't think it can be published on Nature Communications at this stage, my major concerns are listed below:

1. It is interesting that the response at left edge and right edge are different. But why this difference corresponds with BPVE? Why the authors define $IBPVE = \pm(I_{ph}(xL)-I_{ph}(xR))/2$? Besides, the authors assume the device geometry is perfectly symmetric, while this assumption seems contrary to the STEM image in figure 2 d&f.
2. In figure 2j, the polarization dependent photocurrent at position A, B and C are also puzzling. Why the polarities of photocurrent vary with polarization at position B, while the responses at position C are negative and polarization independent? I recommend to add a scanning photocurrent mapping data for the device in figure 2.
3. Could the authors explain why the non-local photocurrent measurement is less influenced by extrinsic photovoltaic effects?
4. In page 9, the authors say "The strong symmetry breaking in edge-embedded structures enables photo-excited charges possessing a finite momentum along y-direction during the charge transfer process." Why the charge transfer process which is mainly in x-z direction gives a momentum along y-direction?
5. I recommend a scanning photocurrent spectroscopy for the WS₂/ReS₂ heterostructure to see what happens at the heterostructure.

There are also some errors to correct in figures. For example, in the caption of Figure 2, "c and d show the enlarged regions of left and right edge-embedded structures" should be correct to "d and f show the enlarged regions of left and right edge-embedded structures".

Reviewer #2 (Remarks to the Author):

This manuscript mainly reported the observation of photocurrent at the artificially engineered 1D edges in stacked vdW materials. The authors attributed the observed photoelectric response to bulk photovoltaic effect (BPVE) by experiments with spatial, light polarization, and power dependance of photocurrents. Although the authors showed the potential universality of the zero-bias photocurrent in edge-embedded vdW structures, in my opinion, the smoking gun to prove its origin from BPVE is still lacking. Therefore, before the consideration of this work for publication in Nature Communications, several issues need to be carefully clarified for the better understandings on the inherent physical origin.

1. In Figure 1, the authors summarized the strategies of inversion symmetry breaking induce BPVE in low-dimensional systems and showed that in vdW heterostructures the top-layer bending at edges should induce strong symmetry breaking and thus leads to strong BPVE. It seems to be discrepant with or with unclear relations to the results as presented in Figure 2 and 3. Firstly, since the bending of the edge-embedded vdW structures is pointing to the out-of-plane direction, strain gradients mainly lie in the z direction and the in-plane x direction. It is straightforward that significant BPVE can be observed in strained directions as the results presented in earlier works (Nat. Nanotech. 16, 894–901, 2021; 18, Nat. Nanotech.18, 36–41 (2023)). But in the devices, photocurrents stem from the y direction. Secondly, in the device, the photocurrent collecting electrodes were deposited on the bottom layer, or in other words, the deformed structure in the top layer actually was not involved directly in the circuits.

Therefore, the origin of the observed photocurrent is not clear and is only mentioned in the discussion part with potential charge transfer behavior at the interface.

2. It is quite puzzled that the photocurrents from the left and right edges in all devices are reverse in both the local and non-local measurements. I would like to suggest the authors to elaborate on this point.
3. To get the BPVE-induced photocurrent, the authors used an average way by the subtraction of the photocurrents from left and right edges. It is very strange that photocurrent peak from nonlocal measurement in Figure 2h is centered at different sample position to that from local measurement in Supplementary Figure 1b. Can the authors explain this point?
4. In Figure 4b and 4d, the noise-level photocurrents seem to show no differences for the readers. Clearer results should be presented to support their arguments.
5. It seems there is the lack of context with respect to earlier studies on the BPVE of vdW materials. Recent advances on this research area should be paid more attentions in this work.

Reviewer #3 (Remarks to the Author):

Liang et.al. reported experimental observation of strong bulk photovoltaic effect in engineered edge-embedded van der Waals structures. Edges or surfaces are certainly the active places to look for photocurrent responses due to the lowered symmetry. The observation of contrasting edge photocurrent signals is interesting. But the study is a bit immature at this stage and needs to be significantly improved before I can make my recommendation. Below I'd like to explain my concerns.

1. My main concern is about the punchline of the study. I don't know the key message the authors want to deliver. What is the punchline here in the edge-embedded system? If they want to emphasize the strong response, they need to compare the response with p-n junctions or the BPVE in other systems. Currently, the magnitude discussion is very limited (if any).

On the other hand, if the authors want to emphasize the mechanism, they need to make more effort to understand it. To confirm the bulk photovoltaic nature of this response, one needs to do a symmetry analysis for the lattice structure, similar to those performed in the literature. Particularly, I want to draw attention to <https://www.nature.com/articles/s41467-019-13713-1.pdf>, which the authors cited as Ref.[39]. There, opposite photocurrents from edges were also observed. The authors should do a similar symmetry analysis and relate it to their observed polarization dependence. Why the two edges show opposite response is unclear without symmetry analysis.

2. ReS₂: ReS₂ has a very low symmetry compared to MoS₂ and hBN. It has strong anisotropy in the plane, which might be relevant to polarization-dependent measurements. In Fig. 2b, the schematic doesn't seem to be ReS₂.

3. Polarization dependence: The polarization dependence of ReS₂/ReS₂ is nice. I'd like to know the polarization dependence of the other structures, MoS₂/MoS₂, MoS₂/ReS₂, etc. For ReS₂/ReS₂ structure, due to the high anisotropic structure, it is not surprising to expect strong polarization dependence. It shouldn't be difficult to know the crystallographic axes of ReS₂ (by Raman or even just the optical image) and relate the polarization with the crystal structure to understand the mechanism better.

4. "Nonlocal" photocurrent: When the authors did the nonlocal photocurrent measurement with electrodes 2 and 3 in Fig. 2a, was electrode 1 floating or grounded? I don't understand why they can measure a signal if 1 is grounded. On the other hand, the authors should explain more about why this scheme can help exclude extrinsic effects and exactly what extrinsic effects they have in mind.

Related to the above, when they do hBN/ReS₂ or ReS₂/hBN, they only show the local measurement; why is that? It is not a fair comparison. Likely, hBN/ReS₂ or ReS₂/hBN does not show similar signals because of the different geometry and measurement schemes.

**Point-by-Point Response to Reviewers' Comments for
Manuscript (NCOMMS-23-01865)**

We highly appreciate the editors and reviewers for the precious time and effort in reviewing our paper and providing constructive suggestions. Your insightful comments lead to considerable improvement in the current version. We have carefully considered and incorporated all of your valuable comments in the revised manuscript. In this letter, we would like to respond to all your comments in a point-by-point manner.

Overviews of major changes

We have made the following important changes in the current version to address your main comments and suggestions.

Firstly, we have done symmetry analysis and the first-principles calculations on vdW edges and edge-embedded structures (see **Box 1** below). The origin of BPVE and reversed polarity of BPVE-induced photocurrents at left and right edge-embedded structures can be well explained.

Secondly, better and new samples have been made and characterized. The quality of data has been significantly improved. Besides, all data in the main text are now from non-local measurements. The advantages of non-local measurements compared with local measurements have been clearly explained.

Thirdly, we have put more attention on recent advances on BPVE in vdW materials. We have made clearer statements on the novelty of the proposed edge-embedded structures.

Finally, we have made every effort to fully address all of your other concerns and suggestions.

Box 1. Symmetry analysis and BPVE origin in ReS₂ edge-embedded structures

Without considering any extrinsic effect, when light (with electric field vector \vec{E} and wave vector \vec{q}) shines on materials, a DC current density can be generated which can be expressed as

$$J_1^{\text{DC}} = \sigma_{\text{ijk}}^{(2)}(w, \vec{q}) E_j E_k^*, \quad (1)$$

where $\sigma_{\text{ijk}}^{(2)}(w, \vec{q})$ is the second-order nonlinear tensor governed by the geometric symmetry of crystalline structure.

If we only consider the \vec{q} -independent and linear polarization-dependent term, we have

$$J_1^{\text{LBPVE}} = \frac{1}{2} \sum_{j,k} \chi_{\text{ijk}} (E_j E_k^* + E_k E_j^*). \quad (2)$$

Here, J_1^{LBPVE} is the BPVE-induced current density under linear polarized light along l -direction (l represents x -, y -, or z -direction), and $\chi_{\text{ijk}} = \sigma_{\text{ijk}}^{(2)}(w, 0)$. For incident light along z -direction, we have $E_z=0$. Then we have

$$\overline{J^{\text{LBPVE}}} = \begin{bmatrix} \chi_{\text{xxx}} |E_x|^2 + \chi_{\text{xyy}} |E_y|^2 + \chi_{\text{xyx}} E_x E_y^* + \chi_{\text{xyy}} E_y E_x^* \\ \chi_{\text{yxx}} |E_x|^2 + \chi_{\text{yyy}} |E_y|^2 + \chi_{\text{yxy}} E_x E_y^* + \chi_{\text{yxy}} E_y E_x^* \\ \chi_{\text{zxx}} |E_x|^2 + \chi_{\text{zyy}} |E_y|^2 + \chi_{\text{zxy}} E_x E_y^* + \chi_{\text{zyx}} E_y E_x^* \end{bmatrix}. \quad (3)$$

Because current collection direction is along y -direction, we mainly focus on J_y^{LBPVE} :

$$J_y^{\text{LBPVE}} = \chi_{\text{yxx}} |E_x|^2 + \chi_{\text{yyy}} |E_y|^2 + \chi_{\text{yxy}} E_x E_y^* + \chi_{\text{yxy}} E_y E_x^*. \quad (4)$$

For materials with inversion symmetry, when we do inversion transformation ($x, y, z \rightarrow -x, -y, -z$), $\overline{J^{\text{LBPVE}}}(x, y, z) = -\overline{J^{\text{LBPVE}}}(-x, -y, -z)$. χ_{ijk} must be zero. Hence, no BPVE effect can be observed in systems with inversion symmetry. In the following section, we will discuss three situations: 1) pure edges with mirror symmetry; 2) pure ReS₂ edges; 3) ReS₂ edge-embedded structures.

1) Pure edges with mirror symmetry

We take $\langle 100 \rangle$ -directional edge of WTe₂ (see **Fig. R1** below, reproduced from the reference *Nat. Commun.* 10, 5736, 2019 in main text) as an example. A mirror

symmetry plane M_a exists in $\langle 100 \rangle$ -directional edge. Under mirror symmetry ($x, y, z \rightarrow x, -y, z$), $J_y^{\text{LBPVE}}(x, y, z) = -J_y^{\text{LBPVE}}(x, -y, z)$, which makes $\chi_{yxx} = \chi_{yyy} = 0$. For linear polarized light $\vec{E} = [E_0 \cos \alpha, E_0 \sin \alpha, 0]$, where α is the angle between x -direction and light polarization direction. Then we have

$$J_y^{\text{LBPVE}} = \chi_{yxy} E_0^2 \sin 2\alpha. \quad (5)$$

Figure R1. WTe₂ crystalline structure reproduced from Reference *Nat. Commun.* 10, 5736, 2019.

Hence, from the theoretical perspective, there should be a y -directional BPVE current along the edge when light polarization is not along x - and y - directions. The BPVE current vanishes when light is polarized along x - and y - directions. However, previous experiments on WTe₂ $\langle 100 \rangle$ -edges (*Nat. Commun.* 10, 5736, 2019) did not observe detectable J_y^{LBPVE} . This is probably due to the small value of χ_{yxy} in WTe₂.

2) Pure ReS₂ edges

We will focus on edges along Re-chain direction. As shown in **Fig. R2**, no mirror symmetry or rotation symmetry exist in ReS₂ edges. Hence, J_y^{LBPVE} along ReS₂ edge can be expressed as

$$J_y^{\text{LBPVE}} = 2\chi_{yxx} E_0^2 \cos^2 \alpha + 2\chi_{yyy} E_0^2 \sin^2 \alpha + 2\chi_{yxy} E_0^2 \sin 2\alpha. \quad (6)$$

We rewrite equation (6) to

$$J_y^{\text{LBPVE}} = (\chi_{yxx} + \chi_{yyy}) E_0^2 + (\chi_{yxx} - \chi_{yyy}) E_0^2 \cos 2\alpha + 2\chi_{yxy} E_0^2 \sin 2\alpha. \quad (7)$$

Here, the first term $(\chi_{yxx} + \chi_{yyy}) E_0^2$ is polarization-independent, and second and third terms are polarization-dependent. From the theoretical perspective, there should be a y -

directional BPVE photocurrent along pure ReS₂ edges. However, we did not observe obvious signal along ReS₂ edges experimentally, probably due to the small value of χ_{ijk} and narrow region of edge. On the other hand, the inversion symmetry is still approximately preserved in unit cell at edges (see **Fig. R2b**). It could probably significantly reduce the BPVE signal.

Figure R2. **a**, STEM image of ReS₂ crystal (top view). **b,c**, Schematic top (**b**) and cross-sectional (**c**) view of the ReS₂ crystalline structure.

3) ReS₂ edge-embedded structures

In ReS₂ edge-embedded structures, top ReS₂ layers are coupled with bottom ReS₂ edges. Regions near ReS₂ edges should show distinct properties from inner regions. The laser wavelength 532nm is also much larger than the edge region. Thus, we can treat edge-embedded regions as a whole as shown by dashed-line-enclosed region in **Fig. R3** below. Besides, strains also exist in edge-embedded structures in *x*- and *z*- directions. Hence, compared with pure ReS₂ edges, the geometric symmetry of ReS₂ edge-embedded structures is further lowered. The BPVE-induced photocurrent can be

expressed as $J_y^{\text{LBPVE}} = D + C \cos(2\alpha + \varphi)$, where D is the polarization-independent term, C is a constant, and φ is the phase. The lowered symmetry and different geometry configuration of ReS₂ edge-embedded structures could probably host detectable value of D and C .

Figure R3. Cross-sectional schematic of ReS₂ edge-embedded structure.

On the other hand, charge transfers inside ReS₂ edge-embedded structures can further enhance the BPVE photocurrent along edges. To support this, we have performed the first-principles calculations on band structures of ReS₂, ReS₂ edges, and strained ReS₂.

Fig. R4 shows the band structure and density of states (DOS) of a ReS₂ ribbon with edges along Re-chain direction. An energy band, generated by edge states, appears inside the band gap of ReS₂. For top ReS₂ layers, there are strains along x - and z -directions near edge regions. Hence, we further calculated the band structure of bent ReS₂. As shown in **Fig. R5**, at lowest position of conduction band (Γ point, marked by the blue circle), electron states are mostly distributed near bent regions. This indicates that photo-carriers generated in top ReS₂ layers tend to gather in bent regions.

Figure R4. **a**, Band structures of ReS₂ ribbon with edges along Re-chain direction. **b**, Electron states at lowest conductance band marked by the blue circle in **a**. **c**, Density of states (DOS) of the ReS₂ ribbon.

Figure R5. **a**, Band structures of bent ReS₂ flake. Strains are along x - and z -directions **b**, Distribution of electron states at lowest position of conduction band (denoted by the blue circle in **a**) in bent ReS₂ flake.

We compared the DOS of ReS₂ edges and bent ReS₂ near the band edge. As shown in **Fig. R6**, inside edge-embedded structures charge transfers are enabled from top bent ReS₂ to bottom ReS₂ edges. Although electrodes are fabricated on bottom ReS₂, charge transfers allow us to detect BPVE-generated carriers from top ReS₂.

Finally, we compared the results ReS₂/ReS₂ and hBN/ReS₂ structures (see **Fig. R7**). In hBN/ReS₂ structures, no peak/valley features are observed in edge regions. This is because insulating materials, such as hBN, do not absorb light. Hence, the hBN/ReS₂ edge region is merely a pure edge with a different dielectric environment as illustrated in **Fig. R7** below.

Figure R6. DOS of ReS₂ edges and bent ReS₂ near the band edge. Charge transfers are enabled from top bent ReS₂ to bottom ReS₂ edges.

Figure R7. **a**, Schematic of ReS₂/ReS₂ edge-embedded structures. Because light wavelength is much larger than the thickness of ReS₂ and edge region, we can treat bottom edge and top flake near edge as a whole structure. **b**, Schematic of hBN/ReS₂ structure. Since insulating materials, such as hBN, do not absorb light, the hBN/ReS₂ edge region is merely a pure edge with a different dielectric environment.

In conclusion, 1) Through vdW coupling between bottom edge and top flake, we engineered an edge-embedded structure which has low lattice symmetry and distinct local properties. 2) Geometric symmetry is further lowered in ReS₂ edge-embedded structures which could enhance BPVE. Photo-excited carriers in ReS₂ edge-embedded regions will acquire *y*-directional momentum under light excitation. 3) ReS₂ edges have different band structures from bent and intrinsic ReS₂. Inside edge-embedded structures, photo-excited carriers (with *y*-directional

momentum) tend to accumulated at bottom ReS₂ edges. This further enhance the y-directional photocurrents along edges.

In the revised manuscript, above discussions and figures have been incorporated into the main text and supplementary materials.

Point-by-point response to referee 1

We would like to thank you for your thoughtful comments and suggestions. We truly appreciate the time and efforts you invested in our paper. With your help, our paper has been improved substantially. In this letter, we will respond to each of your comments following the order in which they appear in your report.

In the manuscript, Liang et al. reported the experimental observation of non-local photocurrent response at the edge of ReS₂/ReS₂ homo-structure as well as several other vdWs structures. They attribute this to a strong bulk photovoltaic effect (BPVE) in the edge-embedded vdWs structures. BPVE has drawn lots of research interests in the past few years in order to find new candidates for optoelectronics applications. The experimental results are interesting. However, I think the experimental data cannot support that the photocurrent at the nano edge embedded in assembled VdW structure comes from BPVE. So I don't think it can be published on Nature Communications at this stage, my major concerns are listed below:

1. It is interesting that the response at left edge and right edge are different. But why this difference corresponds with BPVE? Why the authors define $IBPVE = \pm(I_{ph}(xL) - I_{ph}(xR))/2$? Besides, the authors assume the device geometry is perfectly symmetric, while this assumption seems contrary to the STEM image in figure 2 d&f.

Response: Thank you for pointing out these important issues to us. We agree with you that left and right edges are not perfectly symmetric as shown by the STEM. It is also impossible to find two identical edges in any samples. Hence, it is reasonable that photoresponses at left and right edges are different. If only extrinsic photovoltaic effect exists at two edges, such as photothermoelectric and built-in pn junction effect, the trend of photocurrents at two edges should be the same although the amplitude may be different, as illustrated in **Fig. R8a**. However, our results show that both amplitude and trend are totally different at edge-embedded positions (see **Fig. R8b**). Position-dependent photocurrents at one edge bend upwards while photocurrents at the other edge bend downwards. Besides, in many samples, the photocurrent polarity at two edges is reversed with positive value at one edge and negative value at the other edge.

These unique phenomena are not characteristics of extrinsic photovoltaic effect.

Figure R8. **a**, Since left (x_L) and right (x_R) edges are not perfectly the same, difference of photocurrents between left and right edges cannot be simply attributed to BPVE. **b**, Only when remarkable peak and valley features are observed, we are of the opinion that BPVE occurs at two edges.

We attribute these unique phenomena at edge-embedded structures to BPVE due to following reasons. 1) BPVE is a second-order nonlinear optical response arising from the broken inversion symmetry of crystalline structures of materials. ReS₂ edge-embedded regions are low-symmetric regions with broken inversion symmetry, which allows the existence of BPVE. Furthermore, the low symmetry of ReS₂ edge-embedded structures allow a finite photocurrent traveling along y -direction near edges. Please see detailed symmetry analysis and theoretical calculations on ReS₂ edges in **Box 1**. 2) The physical properties of edge-embedded structure are different from those of bulk, since 1D edge states and strains exist in this local region. From this point of view, we can treat the engineered edge-embedded structure as quasi-1D, as schematically shown by dashed-line-enclosed regions in **Fig. R9**. Since a 180° rotation of left edge along z -axis reproduces the structure of right edge, the orientation of left and right edges must be reversed. Hence, the direction of BPVE photocurrents should also be opposite at left and right edges restricted by the symmetry ($J_y^{\text{LBPVE}} \rightarrow -J_y^{\text{LBPVE}}$ under 180° rotation along z -axis), showing good agreement with our experimental results. 3) BPVE effect strongly depends on the polarization of incident light (see analysis in **Box 1** and response to your next comment). Such polarization dependence is observed in edge-

embedded regions. Although we could not exclude all other possibilities, BPVE is the most logical and feasible explanation on our experimental observations at current stage.

Figure R9. Schematic of the edge-embedded structure. The BPVE-induced photocurrents have reversed polarity at left and right edges due to their reversed orientations. Symbols “ \odot ” and “ \otimes ” represents the y and $-y$ direction of BPVE-induced photocurrents I_{BPVE} , respectively.

In this work, $I_{BPVE} = \pm(I_{ph}(x_L) - I_{ph}(x_R))/2$ is denoted as the **estimated** BPVE-induced photocurrent. Since the radius of laser spot is $\sim 2.5 \mu\text{m}$ which is much wider than the effective edge-embedded region, photocurrents from extrinsic effects (photothermoelectric and pn junction effect) are mixed with BPVE-induced photocurrents. Hence, total photocurrents at left and right edges can be written as $I_{ph}(x_L) = I_{BPVE}(x_L) + I_{Extrinsic}(x_L)$ and $I_{ph}(x_R) = I_{BPVE}(x_R) + I_{Extrinsic}(x_R)$, respectively. Then we get $(I_{BPVE}(x_L) - I_{BPVE}(x_R))/2 = (I_{ph}(x_L) - I_{ph}(x_R))/2 - (I_{Extrinsic}(x_L) - I_{Extrinsic}(x_R))/2$. Because extrinsic contribution $I_{Extrinsic}(x_L)$ and $I_{Extrinsic}(x_R)$ are unknown, we are unable to get the BPVE-induced photocurrents unless making assumption that left and right edges are symmetric (with $(I_{Extrinsic}(x_L) = I_{Extrinsic}(x_R))$ and $I_{BPVE}(x_L) = -I_{BPVE}(x_R)$). In our experimental design, we have tried our best to minimize the asymmetric effect through choosing samples with parallel edges and fabricating electrodes perpendicular to edges. The estimated BPVE-induced photocurrent $I_{BPVE} \approx \pm(I_{ph}(x_L) - I_{ph}(x_R))/2$ should be at the same order with the real value.

In the revised manuscript, we made following changes accordingly. More explanation and discussion on position-dependent photocurrents at two edges have been highlighted

in Page 7 and 8 in the main text and Supplementary Fig. 7. We have made clearer statements on the extraction method of BPVE-induced photocurrent I_{BPVE} in Page 7 and 8 in the main text. Brief discussion on symmetry analysis on edges and edge-embedded structures have been incorporated into main text (see Page 8-10). Detailed symmetry analysis is shown in Supplementary Note 1 and Supplementary Fig. 1 and 2. Brief results of the first-principles calculations are shown in Page 12 in the main text. Detailed results of the first-principles calculations are shown in Supplementary Note 1 and Supplementary Fig. 1 and 2.

2. In figure 2j, the polarization dependent photocurrent at position A, B and C are also puzzling. Why the polarities of photocurrent vary with polarization at position B, while the responses at position C are negative and polarization independent? I recommend to add a scanning photocurrent mapping data for the device in figure 2.

Response: Thank you for the valuable suggestion. As you suggested, we plot the photocurrent mapping of the ReS₂/ReS₂ device under 532 nm laser excitation with laser polarization direction along y -direction (see **Fig. R10c**). As shown in **Fig. R10b** and **R10c**, the photocurrent around position C is indeed negative. This is because position C (near $x=0$) is away from two edges. The negative photocurrent comes from the contribution of extrinsic effect, such as photothermoelectric effect. The polarization ratio at position C is relatively weak which is probably due to the special crystalline orientation of top and bottom ReS₂ flakes. For the ReS₂/ReS₂ homostructure sample, the Re-chain directions of bottom and top ReS₂ flakes are along y - and x - directions, respectively, demonstrated by cross-sectional STEM and Raman measurements (see **Fig. R10d**). The 90° difference of crystalline orientation and charge transfers between top and bottom ReS₂ flakes might decrease the polarization ratio at homostructure region.

At position B, the polarity of total photocurrent can vary with the light polarization. As

shown in **Box 1**, the total photocurrent density J_{total} at edge-embedded structures can be expressed as

$$J_{\text{total}} = J_y^{\text{LBPVE}} + J_y^{\text{Extrinsic}} = D + C \cos(2\alpha + \varphi) + J_y^{\text{Extrinsic}}. \quad (8)$$

Here, the first two terms are BPVE-induced photocurrents, where D is the polarization-independent term, $C \cos(2\alpha + \varphi)$ is the polarization-dependent term. In this sample, photocurrent from extrinsic effect $J_y^{\text{Extrinsic}}$ is negative (for example, the negative photocurrent at position C as discussed above). Hence, it is possible that $D + J_y^{\text{Extrinsic}} - C < 0$. When this relation is satisfied, the polarity of total photocurrent can vary with light polarization. This polarity switching phenomenon was also observed in previous BPVE studies in twisted bilayer graphene and Weyl semimetals. (*Nature* 604, 266, 2022; *Nat. Mater.* 18, 471, 2019).

Figure R10. **a**, Optical image of the ReS₂/ReS₂ homostructure device. **b**, Non-local photocurrent along x -direction at $y=20 \mu\text{m}$. The value at position C is negative. **c**, Photocurrent mapping of the ReS₂/ReS₂ homostructure device. Red dashed line enclosed area denotes the ReS₂/ReS₂ homostructure region. **d**, Polarization-dependent Raman measurements of A_{1g} mode of top and bottom ReS₂ flakes.

In the revised manuscript, discussions on polarization-dependent photocurrents have been highlighted in Page 9 and 10 in the main text. The scanning photocurrent image has been included to Supplementary Fig. 6. Polarization-dependent Raman results have been included to Supplementary Fig. 8.

3. Could the authors explain why the non-local photocurrent measurement is less influenced by extrinsic photovoltaic effects?

Response: Thank you for the comment. We should have shown more information about the non-local measurement. As schematically shown in **Fig. R11**, the photocurrents from extrinsic photovoltaic effect $I_{\text{Extrinsic}}$ show different shapes for local and non-local measurements. For local measurement, the target channel is located between drain and source electrodes. Due to the symmetry of device, $I_{\text{Extrinsic}}$ will change direction and show many peak/valley features when laser spot moves from drain to source (see **Fig. R11a**). For non-local measurement, the target channel is located outside drain and source electrodes. Photon-generated carriers will diffuse to electrodes. Closer to electrodes, easier for carriers to reach electrodes. Thus, $|I_{\text{Extrinsic}}|$ will monotonously increase when approaching electrodes (see **Fig. R11b**). The simpler shape of $I_{\text{Extrinsic}} \sim y$ in non-local measurement is preferred to resolve the peak/valley features of BPVE.

Figure R11. Schematics of local and non-local measurements. The simpler shape of $I_{\text{Extrinsic}} \sim y$ in non-local measurement is preferred to resolve the peak/valley features of BPVE.

In the revised manuscript, detailed comparison between local and non-local measurements are shown in Supplementary Note 2. **Fig. R11** has also been included in Supplementary Information (see Supplementary Fig. 3).

4. In page 9, the authors say “The strong symmetry breaking in edge-embedded structures enables photo-excited charges possessing a finite momentum along y -direction during the charge transfer process.” Why the charge transfer process which is mainly in x - z direction gives a momentum along y -direction?

Response: Thank you for pointing out this unclear statement in previous manuscript.

To address this issue, we further performed theoretical calculation of the band structure of ReS_2 edges and symmetry analysis on ReS_2 edge-embedded structures, which are detailly shown in **Box 1**. Firstly, the symmetry analysis suggests that photo-excited charges in edge-embedded structures possess y -directional momentum due to the broken inversion symmetry. Secondly, the first-principles calculations show that inside edge-embedded structures charges tend to transfer from top ReS_2 layers to bottom ReS_2 edges (see **Fig. R12**) since edge region has a lower conductance band energy. The

charge transfer processes in x - and z - directions do not affect the y -directional momentum of charges arising from BPVE in edge-embedded structures.

Figure R12. DOS of ReS₂ edges and bent ReS₂ near the band edge. Because the conductance band edge is lower in ReS₂ edges, charge transfers are enabled from top bent ReS₂ to bottom ReS₂ edges.

In the revised manuscript, we have made clearer statement on charge transfers inside edge-embedded structures as highlighted in Page 12 and 13 in the main text. **Fig. R12** has been added to Fig. 4 in the main text.

5. *I recommend a scanning photocurrent spectroscopy for the WS₂/ReS₂ heterostructure to see what happens at the heterostructure.*

Response: Thank you for the valuable suggestions. The scanning photocurrent spectroscopy of the WS₂/ReS₂ heterostructure is shown in **Fig. R13a** below. $I_{\text{ph}} \sim x$ relationship across two edge-embedded structures (along black dashed line in **Fig. R13a**) is further shown in **Fig. R13b**. Valley/peak features are reproduced in this heterostructure sample. However, we notice that the peak feature is not exactly located at edge-embedded regions. As schematically shown in **Fig. R13c**, the measured total photocurrent is the superposition of I_{BPVE} and $I_{\text{Extrinsic}}$, which could probably lead to asymmetry and shift of peak/valley positions. Besides, the left and right edges are not

perfectly symmetric. Hence, we think the asymmetric feature of measured I_{ph} in this device is reasonable.

Figure R13. **a**, Scanning photocurrent spectroscopy of the WS₂/ReS₂ heterostructure device. **b**, Non-local photocurrent along x -direction at $y=15 \mu\text{m}$. **c**, Asymmetric feature of I_{ph} can be attributed to the superposition of I_{BPVE} and $I_{Extrinsic}$.

In the revised manuscript, we have included **Fig. R13** in Supplementary Information (see Supplementary Fig. 11).

There are also some errors to correct in figures. For example, in the caption of Figure 2, “c and d show the enlarged regions of left and right edge-embedded structures” should be correct to “d and f show the enlarged regions of left and right edge-embedded structures”.

Response: Thank you for pointing out these typos. We have corrected them in the revised manuscript.

Point-by-point response to referee 2

We would like to thank you for your thoughtful comments and suggestions. We truly appreciate the time and efforts you invested in our paper. With your help, our paper has been improved substantially. In this letter, we will respond to each of your comments following the order in which they appear in your report.

This manuscript mainly reported the observation of photocurrent at the artificially engineered 1D edges in stacked vdW materials. The authors attributed the observed photoelectric response to bulk photovoltaic effect (BPVE) by experiments with spatial, light polarization, and power dependence of photocurrents. Although the authors showed the potential universality of the zero-bias photocurrent in edge-embedded vdW structures, in my opinion, the smoking gun to prove its origin from BPVE is still lacking. Therefore, before the consideration of this work for publication in Nature Communications, several issues need to be carefully clarified for the better understandings on the inherent physical origin.

1. In Figure 1, the authors summarized the strategies of inversion symmetry breaking induce BPVE in low-dimensional systems and showed that in vdW heterostructures the top-layer bending at edges should induce strong symmetry breaking and thus leads to strong BPVE. It seems to be discrepant with or with unclear relations to the results as presented in Figure 2 and 3. Firstly, since the bending of the edge-embedded vdW structures is pointing to the out-of-plane direction, strain gradients mainly lie in the z direction and the in-plane x direction. It is straightforward that significant BPVE can be observed in strained directions as the results presented in earlier works (Nat. Nanotech. 16, 894–901, 2021; Nat. Nanotech.18, 36–41 (2023)). But in the devices, photocurrents stem from the y direction. Secondly, in the device, the photocurrent collecting electrodes were deposited on the bottom layer, or in other words, the deformed structure in the top layer actually was not involved directly in the circuits. Therefore, the origin of the observed photocurrent is not clear and is only mentioned in the discussion part with potential charge transfer behavior at the interface.

Response: Thank you for the valuable suggestions! We totally agree with you that significant BPVE can be observed in strained directions. *Nat. Nanotech.* 16, 894–901, 2021 reports that the strain gradient (non-uniform strain in x -direction) in 2H-MoS₂ induces a large BPVE photocurrent along the strain gradient direction. For y -direction

perpendicular to strain gradient, BPVE-induced photocurrent follows $\sin(2\alpha)$ dependence, where α is the angle between strain gradient and light polarization direction. Hence, y -directional photocurrent should vanish when light polarization is along y -direction (perpendicular to strain gradient direction). *Nat. Nanotech.* 18, 36–41, 2023 shows that a uniform strain in 3R-MoS₂ can induce an electronic polarization along the armchair direction regardless the direction of strain. Hence, the BPVE photocurrent is always along the armchair direction of 3R-MoS₂. In our work, a strong photocurrent is observed along the y -direction of edge-embedded structures (perpendicular to the strain direction in top flake), suggesting a different origin of BPVE photocurrent. Hence, taking bottom edge and top flake separately could not explain our observation.

Taking bottom edge and top flake region nearby as a whole structure can well explain our observation. Here, the edge-embedded structure is regarded to the nearby region of bottom edge and top flake as schematically shown in **Fig. R14** below. The physical properties of edge-embedded structure are different from those of bulk, since 1D edge states and strains exist in this local region. From this point of view, we can treat the engineered edge-embedded structure as quasi-1D. To clarify the physical origin of BPVE in edge-embedded structures, we performed symmetry analysis and the first-principles calculations. Detailed results are shown in **Box 1**. Here, we summarized key results of the theoretical analysis: 1) In ReS₂ edge-embedded structures, inversion symmetry and mirror plane (in x - z plane) does not exist. This allows a polarization-independent term D in y -directional BPVE photocurrent density $J_y^{\text{LBPVE}} = D + C \cos(2\alpha + \varphi)$ under polarized light, which agrees with our observations. 2) The geometry configuration of ReS₂ edge-embedded structures is different with pure ReS₂ edges, and crystalline symmetry in ReS₂ edge-embedded structures is further lowered, which could probably enhance the amplitude of BPVE photocurrent with

detectable value of D and C . 3) As shown in **Fig. R14d** below, orientations of left and right edge-embedded structures are reversed resulting in reversed BPVE photocurrents at two edges. This also agrees with our experimental observations. 4) ReS₂ edges have different band structures from bent and intrinsic ReS₂. Inside edge-embedded structures, photo-carriers (with y -directional momentum) tend to transfer from top bent regions to bottom ReS₂ edges as supported by the first-principles calculations (see **Fig. R15** below and **Box 1**). Although electrodes are on bottom ReS₂, charge transfers inside edge-embedded structures allow us to detect BPVE-induced photo-carriers from top ReS₂. This further enhances the BPVE photocurrents along edges.

Figure R14. **a**, Cross-sectional scanning transmission electron microscope image (STEM) of ReS₂ crystal from top view. **b**, Schematic images of nano edges in mechanically exfoliated van der Waals (vdW) layered materials. **c,d**, Schematic images of edge-embedded vdW homo- or hetero-structures proposed in this work. Effective vdW coupling between bottom edge regions and top vdW materials can result in strong symmetry-breaking and quasi-one-dimensional edge-embedded structures, which host strong BPVE effect. Symbols “ \odot ” and “ \otimes ” represents the y and $-y$ direction of BPVE-induced photocurrents I_{BPVE} , respectively.

Figure R15. DOS of ReS₂ edges and bent ReS₂ near the band edge. Because the conductance band edge is lower in ReS₂ edges, charge transfers are enabled from top bent ReS₂ to bottom ReS₂ edges.

In conclusion, through vdW coupling between bottom edge and top flake, we engineered an edge-embedded structure which has low geometric symmetry and distinct local properties. Under polarized light, photo-excited carriers acquire y -directional momentum governed by the BPVE in edge-embedded structure. Inside edge-embedded structures, photo-excited carriers tend to accumulated in bottom edges through charge transfer process. Besides, the charge transfer process in x - and z - directions does not affect the y -directional momentum of carriers. As a result, we are able to observe a detectable BPVE photocurrent along y -direction in edge-embedded structures.

In the revised manuscript, the origin of BPVE in edge-embedded structures are incorporated into the main text (see highlighted region in Page 8, 9, 12 and 13). Detailed theoretical analysis is shown in Supplementary Note 1, Supplementary Fig. 1 and 2. **Fig. R14** is shown in Fig. 1 in the main text. **Fig. R15** is shown in Fig. 4 in the main text.

2. It is quite puzzled that the photocurrents from the left and right edges in all devices are reverse in both the local and non-local measurements. I would like to suggest the authors to elaborate on this point.

Response: We apologize for the inadequate explanation on the observation of reversed photocurrents at left and right edges. As analyzed in **Box 1**, a y -directional BPVE photocurrent $J_y^{\text{LBPVE}} = D + C \cos(2\alpha + \varphi)$ can be stemmed in edge-embedded regions due to the low crystalline symmetry, where D is the polarization-independent term. As shown in **Fig. R14d** above, the orientations of left and right edge-embedded structures are reversed (with a 180° rotation along z -axis). Since BPVE effect strongly depends on the crystalline structure and orientation of materials, the direction of BPVE photocurrents should also be opposite at left and right edges restricted by the symmetry ($J_y^{\text{LBPVE}} \rightarrow -J_y^{\text{LBPVE}}$ under 180° rotation along z -axis).

In the revised manuscript, we have elaborated on the observation of reversed photocurrents at left and right edges in Page 9 in the main text. We have significantly modified the Fig. 1 in the main text accordingly.

3. To get the BPVE-induced photocurrent, the authors used an average way by the subtraction of the photocurrents from left and right edges. It is very strange that photocurrent peak from nonlocal measurement in Figure 2h is centered at different sample position to that from local measurement in Supplementary Figure 1b. Can the authors explain this point?

Response: This is an insightful observation! The peaks of BPVE-induced photocurrents are indeed at different positions for non-local and local measurements as shown in **Fig. R16** below. We attribute this phenomenon to following reasons. Firstly, for local measurements, the electrodes 1 and 2 are not perfectly symmetric. As a result, the peak may not locate in the middle position of channel (see **Fig. R16b**). Secondly, for non-local measurements, photocurrents are generated when photo-excited carriers diffuse to electrode 2. In real situations, the mean diffusion length ξ is finite. Thus, nonlocal

photocurrent also strongly depends on the distance between laser spot and electrode 2. Shorter distance will result in a larger photocurrent. We can introduce a scaling curve $l(y)$ that describes the diffusion characteristics of carriers. Based on conventional semiconductor theory, $l(y) = \exp(-y/\xi)$. If we assume the non-local photocurrent with and without considering ξ to be $f(y)$ and $g(y)$, respectively, we have $f(y) = \exp(-y/\xi)g(y)$. The scaling term $\exp(-y/\xi)$ will result in a shift of photocurrent peak towards electrode 2 (see **Fig. R16c**).

Figure R16. **a**, Extracted BPVE-induced photocurrent through non-local measurements. **b**, Extracted BPVE-induced photocurrent through local measurements. **c**, The finite mean diffusion length of carriers ξ might leads to the different peak positions in local and non-local measurements.

In the revised manuscript, we have given possible explanation on different peak positions in local and non-local measurements in Supplementary Note 2. **Fig. R16** is included in Supplementary Fig. 5.

4. In Figure 4b and 4d, the noise-level photocurrents seem to show no differences for the readers. Clearer results should be presented to support their arguments.

Response: Thank you for the suggestion. As shown in **Fig. R17** below, we have made new and better hBN/ReS₂ (ReS₂ in bottom) and ReS₂/hBN (hBN in bottom) samples and have performed local and nonlocal measurements. For clarity and better comparison, we put total photocurrents instead of BPVE-induced currents. For both local and nonlocal measurements, photocurrents along left and right edges show the same trend without distinguishable peak/valley signatures. Since insulating materials, such as hBN, do not absorb light, the hBN/ReS₂ edge region is merely a pure edge with a different dielectric environment as illustrated in **Fig. R18** below.

In the revised manuscript, we have included **Fig. R17a,c,d,f** in the main text (see Fig. 4a-d) and have included **Fig. R17b,e** in the Supplementary Fig. 12. **Fig. R18** has been included in Fig. 4e in the main text.

Figure R17. **a-c**, Optical image (**a**), local photocurrent (**b**) and non-local photocurrent (**c**) along two edges in the new hBN/ReS₂ (ReS₂ in bottom) sample. **d-f**, Optical image (**d**), local photocurrent (**e**) and non-local photocurrent (**f**) along two edges in the new ReS₂/hBN (ReS₂ in top) sample. For both local and nonlocal measurements, photocurrents along left and right edges show the same trend without distinguishable

peak/valley signatures.

Figure R18. **a**, Schematic of ReS₂/ReS₂ edge-embedded structures. Because light wavelength is much larger than the thickness of ReS₂ and edge region, we can treat bottom edge and top flake near edge as a whole structure. **b**, Schematic of hBN/ReS₂ structure. Since insulating materials, such as hBN, do not absorb light, the hBN/ReS₂ edge region is merely a pure edge with a different dielectric environment.

5. It seems there is the lack of context with respect to earlier studies on the BPVE of vdW materials. Recent advances on this research area should be paid more attentions in this work.

Response: Following your suggestion, in the revised manuscript, we have rewritten the introduction part and paid more attentions to recent advances on BPVE. We especially paid more attention to earlier works on WTe₂ edges. Please see highlighted region in Page 2 and 3 in the main text. For your convenience, we also copy the revised text below:

“Recent studies suggest that Weyl semimetals and narrow-bandgap semiconductors, such as TaAs and tellurium, can support stronger BPVE²¹⁻²⁵. Other strategies, such as lowering lattice symmetry and dimensionality, were also proposed to effectively enhance BPVE^{3-6, 12, 26-35}. For example, BPVE-induced photocurrent density in one-dimensional (1D) WS₂ nanotube is over 1 Acm⁻², which is among the highest reported values³. Van der Waals (vdW) layered materials, with low dimensionality, rich species and good flexibility, offer another ideal platform to investigate BPVE. Although

majority vdW materials possess lattice inversion symmetry, several approaches have been developed to generate strong BPVE. For example, strain gradient in 2H-MoS₂ and an uniform strain in rhombohedrally stacked MoS₂ (3R-MoS₂) can break the lattice inversion symmetry and induce giant in-plane short-circuit photocurrents under linearly polarized light^{4, 36}; Moiré-pattern-induced symmetry breaking in twisted bilayer graphene allow the observation of pronounced BPVE⁵; Interface of black-phosphorus/WSe₂ heterostructure generates in-plane strong polarization and directional photocurrents in WSe₂ due to the broken inversion symmetry¹²; The spontaneous polarization in out-of-plane direction was also observed in 3R-MoS₂³⁷; Non-centrosymmetric nano-antennas can assist the generation of artificial BPVE in centrosymmetric graphene flakes^{6, 38}; Berry curvature dipole in monolayer topological insulator WTe₂ also support BPVE and can be controlled with a vertical displacement field³⁹. On the other hand, nano edges, where the periodic crystalline structure of a material is interrupted, show broken inversion symmetry and theoretically can host BPVE (see Fig. 1a and 1b)^{21, 40, 41}. Recently, BPVE-induced photocurrents travelling along specific edges were experimentally reported in type-II Weyl semimetal WTe₂²¹. The edge photocurrents strongly depend on the geometric symmetry near edges and are possibly enhanced by fermi-arc type surface states²¹. These results suggest vdW nano edges a promising platform for BPVE investigations. However, for majority vdW materials, including MoS₂ and ReS₂, BPVE in nano edges is either negligible or indistinguishable from extrinsic effects.”

Point-by-point response to referee 3

We would like to thank you for your thoughtful comments and suggestions. We truly appreciate the time and efforts you invested in our paper. With your help, our paper has improved substantially. In this letter, we will respond to each of your comments following the order in which they appear in your report.

Liang et.al. reported experimental observation of strong bulk photovoltaic effect in engineered edge-embedded van der Waals structures. Edges or surfaces are certainly the active places to look for photocurrent responses due to the lowered symmetry. The observation of contrasting edge photocurrent signals is interesting. But the study is a bit immature at this stage and needs to be significantly improved before I can make my recommendation. Below I'd like to explain my concerns.

1. My main concern is about the punchline of the study. I don't know the key message the authors want to deliver. What is the punchline here in the edge-embedded system? If they want to emphasize the strong response, they need to compare the response with p-n junctions or the BPVE in other systems. Currently, the magnitude discussion is very limited (if any). On the other hand, if the authors want to emphasize the mechanism, they need to make more effort to understand it. To confirm the bulk photovoltaic nature of this response, one needs to do a symmetry analysis for the lattice structure, similar to those performed in the literature. Particularly, I want to draw attention to <https://www.nature.com/articles/s41467-019-13713-1.pdf>, which the authors cited as Ref.[39]. There, opposite photocurrents from edges were also observed. The authors should do a similar symmetry analysis and relate it to their observed polarization dependence. Why the two edges show opposite response is unclear without symmetry analysis.

Response: Thank you for the insightful comments and suggestions. We share you concerns that the previous version of the manuscript lacks of symmetry analysis and mechanism explanation. Firstly, we have emphasized the previous work on WTe₂ edges (*Nat. Commun.* 10, 5736, 2019) in introduction part of the revised manuscript, where opposite photocurrents were observed at specific edges. The strong edge photocurrents in WTe₂ are probably related to the fermi-arc type surface states. However, for majority

vdW materials, including MoS₂ and ReS₂, such BPVE-induced photocurrents along edges were rarely observed or reported, although broken inversion symmetry exists near edges. This is probably due to the small value of second-order DC photoconductivity tensor or narrow edge regions influenced by the broken inversion symmetry. In our work, we constructed vdW edge-embedded structures which allows observation of BPVE at edges of ReS₂ and MoS₂. The edge-embedded structure is regarded to the nearby region of bottom edge and top flake as schematically shown in **Fig. R19d** below. The physical properties of edge-embedded structure are different from those of bulk, since quasi-1D edge states and strains exist in this local region. From this point of view, we can treat the edge-embedded structure as quasi-1D. The edge-embedded structure shows lower geometric symmetry and distinct local properties, which could support a strong BPVE. Hence, we believe our strategy is unique and provides useful information for engineering BPVE in various material systems.

Secondly, according to your suggestions, we have performed symmetry analysis and the first-principles calculations on ReS₂ edges and edge-embedded structures. Detailed analysis is shown in **Box 1** in the beginning part of this reply letter. Here, we briefly summarized key results of the theoretical analysis: 1) In ReS₂ edge-embedded structures, inversion symmetry and mirror plane (in x - z plane) does not exist. This allows a polarization-independent term in y -directional BPVE photocurrent $J_y^{\text{BPVE}} = D + C \cos(2\alpha + \varphi)$. 2) The geometry configuration of ReS₂ edge-embedded structures is different with pure ReS₂ edge, and crystalline symmetry in ReS₂ edge-embedded structures is further lowered, which could probably enhance the amplitude of BPVE photocurrent with detectable value of D and C . 3) As shown in **Fig. R19d** below, orientations of left and right edge-embedded structures are reversed resulting in reversed BPVE photocurrents at two edges. This also agrees with our experimental

observations. 4) ReS₂ edges have different band structures from bent and intrinsic ReS₂. Inside edge-embedded structures, photo-carriers (with y -directional momentum) tend to transfer from top bent regions to bottom ReS₂ edges as supported by the first-principles calculations. This further enhances the BPVE photocurrents along edges.

Figure R19. **a**, Cross-sectional scanning transmission electron microscope image (STEM) of ReS₂ crystal from top view. **b**, Schematic images of nano edges in mechanically exfoliated van der Waals (vdW) layered materials. **c,d**, Schematic images of edge-embedded vdW homo- or hetero-structures proposed in this work. Effective vdW coupling between bottom edge regions and top vdW materials can result in strong symmetry-breaking and quasi-one-dimensional edge-embedded structures, which host strong BPVE effect. Symbols “ \odot ” and “ \otimes ” represents the y and $-y$ direction of BPVE-induced photocurrents I_{BPVE} , respectively.

In conclusion, through vdW coupling between bottom edge and top flake, we engineered an edge-embedded structure which has low geometric symmetry and distinct local properties. Under polarized light, photo-excited carriers acquire y -directional momentum governed by the BPVE in edge-embedded structure. Inside edge-embedded structures, photo-excited carriers tend to accumulated in bottom

edges through charge transfer process. Besides, the charge transfer process in x - and z - directions does not affect the y -directional momentum of carriers. As a result, we are able to observe a detectable BPVE photocurrent along y -direction in edge-embedded structures.

In the revised manuscript, we have emphasized the previous work on WTe₂ edges (*Nat. Commun.* 10, 5736, 2019) in introduction part of the main text (see Page 2 and 3). We have also clearly pointed out the uniqueness of our strategy in introduction (see Page 3 and 4). The origin and explanation on BPVE in edge-embedded structures have been incorporated into the main text (see Page 8, 9, 12 and 13). Detailed theoretical analysis is shown in Supplementary Note 1, Supplementary Fig. 1 and 2. **Fig. R19** has been included in Fig. 1 in the main text.

2. ReS₂: ReS₂ has a very low symmetry compared to MoS₂ and hBN. It has strong anisotropy in the plane, which might be relevant to polarization-dependent measurements. In Fig. 2b, the schematic doesn't seem to be ReS₂.

Response: We agree with you that the anisotropic crystalline structure of ReS₂ could influence the polarization-dependent photocurrent measurements. We further performed Raman characterizations (see **Fig. R20c** below). Based on STEM and Raman results, the Re-chain direction of top and bottom ReS₂ are along x - and y -direction respectively (see **Fig. R20**). Electron mobility and optical absorption are higher along the Re-chain direction. In this sample, the maximum photocurrent reaches the maximum value when light polarization is along edges (y -direction), same with the Re-chain direction of bottom ReS₂. The measured polarization dependence of photocurrents should be a mixed effect from both anisotropic crystalline structure and BPVE. We are aware of that it is difficult to separate BPVE from the polarization-dependent measurements. However, compared with homostructure region away from

edge, the polarization dependence is stronger at edge-embedded regions which is probably due to BPVE.

Figure R20. **a**, Optical image of the ReS₂/ReS₂ device. **b,c**, Polarization-dependent Raman spectra of ReS₂ flakes.

Fig. R21 shows polarization-dependent photocurrents at three representative positions A, B, and C in another ReS₂/ReS₂ sample. For bottom and top ReS₂ flakes, Re-chain directions θ are 120° and 90°, respectively, marked by white arrows in **Fig. R21**. At position B away from edges, photocurrent reaches the maximum value at $\theta \approx 120^\circ$ (Re-chain direction). At pure edge (position C), photocurrent reaches the maximum value at $\theta \approx 100^\circ$, which is close to the Re-chain direction. However, at edge-embedded structures (position A), photocurrent reaches the maximum value at $\theta \approx 10^\circ$, which is close to the y-direction (the edge direction). This further demonstrates that BPVE does influence the polarization properties of photocurrent.

Figure R21. **a**, Optical image of another ReS₂/ReS₂ homostructure device. Scale bar is 10 μm . **b-d**, Polarization-dependent photocurrents at three representative positions A (**b**), B (**c**), and C (**d**). At vdW nano-tunnels (point A), photocurrents reach the maximum value when laser is polarized near the y-direction ($\theta \sim 0^\circ$). At points B (the middle region of bottom flake) and C (the edge-embedded region), photocurrents reach the maximum value along Re-chain direction $\theta \sim 120^\circ$.

In the revised manuscript, we have shown more discussions on the polarization-dependent photocurrent results as highlighted in Page 9 and 10 in the main text and in Supplementary Fig. 8 and 9.

Finally, according to your suggestion, the schematic in Fig. 2b has been replaced with the crystalline structure of ReS₂ as shown in **Fig. R22** below.

Figure R22. Schematic of the ReS₂/ReS₂ homostructure device.

3. Polarization dependence: The polarization dependence of ReS₂/ReS₂ is nice. I'd like to know the polarization dependence of the other structures, MoS₂/MoS₂, MoS₂/ReS₂, etc. For ReS₂/ReS₂ structure, due to the high anisotropic structure, it is not surprising to expect strong polarization dependence. It shouldn't be difficult to know the crystallographic axes of ReS₂ (by Raman or even just the optical image) and relate the polarization with the crystal structure to understand the mechanism better.

Response: Thank you for the suggestion. Discussion on polarization-dependent photocurrents in ReS₂/ReS₂ homostructures are shown in Comment 2 above. Since the anisotropic structure do influence the polarization-dependent photocurrents, we agree that it is necessary to show polarization data of isotropic materials, such as MoS₂/MoS₂ homostructure. **Fig. R23** shows the polarization-dependent photocurrents at two representative positions of the MoS₂/MoS₂ homostructure. No polarization dependence is observed outside edge-embedded region which is consistent with the isotropic crystalline structure of MoS₂. However, at edge-embedded region, polarization dependence is observed and the maximum value is along the edge direction. This provides another evidence that BPVE exists at edge-embedded structures.

Figure R23. **a**, Optical image of the MoS₂/MoS₂ homostructure device. **b**, Polarization-dependent photocurrents at edge-embedded structure (orange line) and outside edge-embedded structure (blue line). Positions are denoted by orange and blue circles in **(a)**, respectively.

In the revised manuscript, we have shown the polarization-dependent results of MoS₂/MoS₂ homostructure in the main text (see Page 10 and 11). **Fig. R23b** has also

been included in the main text (see Fig. 3d).

4. “Nonlocal” photocurrent: When the authors did the nonlocal photocurrent measurement with electrodes 2 and 3 in Fig. 2a, was electrode 1 floating or grounded? I don’t understand why they can measure a signal if 1 is grounded. On the other hand, the authors should explain more about why this scheme can help exclude extrinsic effects and exactly what extrinsic effects they have in mind.

Related to the above, when they do hBN/ReS₂ or ReS₂/hBN, they only show the local measurement; why is that? It is not a fair comparison. Likely, hBN/ReS₂ or ReS₂/hBN does not show similar signals because of the different geometry and measurement schemes.

Response: Thank you for pointing out these important issues.

Firstly, for non-local measurements, electrode 1 is always floating. As schematically shown in **Fig. R24** below, high density of carriers will form around the laser spot under laser excitation. Photo-excited carriers will diffuse to electrodes 2 and result in a short-circuit current. As schematically shown in **Fig. R25**, the photocurrents from extrinsic photovoltaic effect $I_{\text{Extrinsic}}$ show different shapes for local and non-local measurements. The main extrinsic effects are the build-in electric field in metal-ReS₂ junction and photothermoelectric effect under thermal gradient generated by laser spots. For local measurement, the target channel is located between drain and source electrodes. Due to the symmetry of device, $I_{\text{Extrinsic}}$ will change direction and show peak/valley features when laser spot moves from drain to source (see **Fig. R25a**). For non-local measurement, the target channel is located outside drain and source electrodes. Photon-generated carriers will diffuse to electrodes. Closer to electrodes, easier for carriers to reach electrodes. Thus, $|I_{\text{Extrinsic}}|$ will monotonously increase when approaching electrodes (see **Fig. R25b**). The simpler shape of $I_{\text{Extrinsic}}$ curve in non-local measurement is preferred to resolve the peak/valley features of BPVE-induced photocurrents.

In the revised manuscript, we have clearly pointed out that electrode is floating in the

main text (see Page 7). Detailed comparison between local and non-local measurements are shown in Supplementary Note 2 and Supplementary Fig. 3.

Figure R24. Schematic of non-local measurement of photocurrents.

Figure R25. Schematics of local and non-local measurements. The simpler shape of $I_{\text{Extrinsic}} \sim y$ in non-local measurement is preferred to resolve the peak/valley features of BPVE.

Secondly, we have made new and better hBN/ReS₂ (ReS₂ in bottom) and ReS₂/hBN (hBN in bottom) samples, and have performed local and non-local measurements (see **Fig. R26** below). For both local and nonlocal measurements, photocurrents along left and right edges show the same trend without distinguishable peak/valley signatures. Since insulating materials, such as hBN, do not absorb light, the hBN/ReS₂ edge region

is merely a pure edge with a different dielectric environment as illustrated in **Fig. R27** below.

In the revised manuscript, we have included **Fig. R26a,c,d,f** in the main text (see Fig. 4a-d), have included **Fig. R26b,e** in the Supplementary Fig. 12 and have included **Fig. R27** in Fig.4.

Figure R26. a-c, Optical image (a), local photocurrent (b) and non-local photocurrent (c) along two edges in the new hBN/ReS₂ (ReS₂ in bottom) sample. d-f, Optical image (d), local photocurrent (e) and non-local photocurrent (f) along two edges in the new ReS₂/hBN (ReS₂ in top) sample.

Figure R27. a, Schematic of ReS₂/ReS₂ edge-embedded structures. Because light wavelength is much larger than the thickness of ReS₂ and edge region, we can treat

bottom edge and top flake near edge as a whole structure. **b**, Schematic of hBN/ReS₂ structure. Since insulating materials, such as hBN, do not absorb light, the hBN/ReS₂ edge region is merely a pure edge with a different dielectric environment.

Concluding remarks:

We would like to thank the entire editorial team once again for your very insightful comments and valuable suggestions, which helped us greatly improve the paper. We hope that we have addressed all of your comments to meet the standard of publication.

REVIEWER COMMENTS

Reviewer #1 (Remarks to the Author):

The revised manuscript is significantly improved over the former version and I think the current version is almost ready for publication in Nature Communications. I have the following two questions to be addressed which may further improve the manuscript:

1. Why the response in figure 2h has a peak at the center?
2. In page 9, the authors used polarization dependent response at position C as a comparison, while the measurement position should be put on the pure edge, not in the bulk. The polarization dependent response in the bulk just support the anisotropic response of the material.

Reviewer #2 (Remarks to the Author):

In the reply, I appreciate the authors' effort on revising the manuscript with more experimental measurements on new samples and the symmetry analysis together with the first-principles calculations. However, the responses on the origin of BVPE in edge-embedded vdW structure are not that convincing enough. In the symmetry analysis for ReS₂ edge-embedded structure, the BPVE induced photocurrent density is expressed by $J_y^{\text{BPVE}} = D + C \cos(2\alpha + \varphi)$. To a certain extent, this description is rather obscure since the author did not mention that how this relation is derived but only mentioned the term D and C are probably detectable. For better understandings, I would like to suggest the authors to make more clear clarifications. Second, the authors emphasis on the treatment of the edge-embedded region as a whole for the consideration of inversion symmetry breaking. The polarity of the BPVE photocurrent at the left and right edges are then fixed to have opposite directions. The authors aimed to provide a universal physical picture or mechanism for the photocurrent in most vdW edge-embedded structures. However, by comparing the results from ReS₂/ReSe₂ and WSe₂/ReSe₂ samples, different signs for x_L and x_R can be found. As the charge transfer process at edges are attributed to enhance the photocurrent, it is necessary for the authors to pay more attention to the details of the photocurrent generation in different 2D materials-based edge-embedded structures since the charge transfer behavior in other 2D materials may be different.

Reviewer #3 (Remarks to the Author):

The authors provided a clear and systematic reply to the reviewers' questions. Those answers are technically reasonable and I support the publication of the current manuscript.

**Point-by-Point Response to Reviewers' Comments for
Manuscript (NCOMMS-23-01865A)**

We highly appreciate the editors and reviewers for the precious time and effort in reviewing our paper and providing constructive suggestions. Your insightful comments lead to considerable improvement in the current version. We have carefully considered and incorporated all of your valuable comments in the revised manuscript. In this letter, we would like to respond to all your comments in a point-by-point manner.

Point-by-point response to referee 1

We would like to thank you for your thoughtful comments and suggestions. We truly appreciate the time and efforts you invested in our paper. With your help, our paper has been improved substantially. In this letter, we will respond to each of your comments following the order in which they appear in your report.

The revised manuscript is significantly improved over the former version and I think the current version is almost ready for publication in Nature Communications. I have the following two questions to be addressed which may further improve the manuscript:

1. Why the response in figure 2h has a peak at the center?

Response: Thank you for the positive assessment of the manuscript. It is true that the BPVE-induced photocurrent I_{BPVE} has peak features, similar to that observed in WS₂ nanotubes (Nature 570, 349-353, 2019). When laser is focused near the boundary of pure-edge/edge-embedded-structure, edge-embedded structures absorb less photons leading to the reduction of photocurrents away from the center. This is the possible reason that I_{BPVE} curve has a peak feature. For the ReS₂/ReS₂ sample in Figure 2h, the peak position of I_{BPVE} is exactly at the center of the edge-embedded structure. We think it is a coincidence due to following reasons. 1) The peak locations are different for

different samples as shown in **Fig. R1a-c**. The peak features are not all at the center. This is because electrodes and contact resistance are not perfectly symmetry and samples are different for different devices. 2) We also observe that the peak position is closer to electrode 2 for non-local measurements compared with local measurements, as shown in **Fig. 2a** and **2b**. The explanation can be found in the 1st Round Response Letter or Supplementary Note 2. For your convenience, we put the explanation here. “For non-local measurements, photocurrents are generated when photo-excited carriers diffuse to electrode 2. In real situations, the mean diffusion length ξ is finite. Thus, nonlocal photocurrent also strongly depends on the distance between laser spot and electrode 2. Shorter distance will result in a larger photocurrent. We can introduce a scaling curve $l(y)$ that describes the diffusion characteristics of carriers. Based on conventional semiconductor theory, $l(y) = \exp(-y / \xi)$. If we assume the non-local photocurrent with and without considering ξ to be $f(y)$ and $g(y)$, respectively, we have $f(y) = \exp(-y / \xi)g(y)$. The scaling term $\exp(-y / \xi)$ will result in a shift of photocurrent peak towards electrode 2 (see **Fig. R2c**).”

Figure R1. a-c, Extracted BPVE-induced photocurrent through non-local measurements for ReS₂/ReS₂ (a), MoS₂/MoS₂ (b), and ReS₂/WS₂ (c).

Figure R2. a,b, Extracted BPVE-induced photocurrent through non-local (a) and local (b) measurements for ReS₂/ReS₂. c, The finite mean diffusion length of carriers ξ might leads to the different peak positions in local and non-local measurements.

From above discussion, we can see that the peak position depends on many factors, including characterization method and symmetry of devices. In the revised manuscript, we have included above discussion (see highlights in Page 8).

2. In page 9, the authors used polarization dependent response at position C as a comparison, while the measurement position should be put on the pure edge, not in the bulk. The polarization dependent response in the bulk just supports the anisotropic response of the materials.

Response: We totally agree with you that polarization dependence at pure edge should be studied. For the ReS₂/ReS₂ sample in Fig. 2, the Re-chain direction (where the photoresponse is maximum) of bottom ReS₂ is occasionally along the edge direction. Hence, this sample is not sufficient enough to show the polarization difference between pure edges and edge-embedded structures. Therefore, in the revised manuscript, we further show another ReS₂/ReS₂ sample with different orientations of pure edge and

edge-embedded structures (see **Fig. R3** below). The Re-chain directions of bottom and top ReS₂ flakes are marked in **Fig. R3a**. As shown in **Fig. 3b-3d**, bulk position (position C) and pure edge (position B) shows similar polarization response with maximum photocurrent near the Re-chain direction ($\theta \sim 30^\circ$), while for edge-embedded structures (position A), the maximum photocurrent response is along the y -direction/edge-direction ($\theta \sim 90^\circ$). This demonstrates that BPVE does influence the polarization dependence at edge-embedded structures.

Figure R3. a, Optical image of another ReS₂/ReS₂ homostructure device. Scale bar is 10 μm . **b-d**, Polarization-dependent photocurrents at three representative positions A (**b**), B (**c**), and C (**d**). At vdW edge-embedded structure (point A), photocurrents reach the maximum value when laser is polarized along y -direction ($\theta \sim -90^\circ$). At points B (the inner region of bottom flake) and C (the edge of bottom flake), photocurrents reach the maximum value along a different angle $\theta \sim 30^\circ$.

In the revised manuscript, we have included Fig. R3 and discussions in the main text (see Fig. 3 and Page 10 and 11).

Point-by-point response to referee 2

We would like to thank you for your thoughtful comments and suggestions. We truly appreciate the time and efforts you invested in our paper. With your help, our paper has been improved substantially. In this letter, we will respond to each of your comments following the order in which they appear in your report.

In the reply, I appreciate the authors' effort on revising the manuscript with more experimental measurements on new samples and the symmetry analysis together with the first-principles calculations. However, the responses on the origin of BPVE in edge-embedded vdW structure are not that convincing enough. In the symmetry analysis for ReS₂ edge-embedded structure, the BPVE induced photocurrent density is expressed by $J_y^{\text{LBPVE}} = D + C \cos(2\alpha + \varphi)$. To a certain extent, this description is rather obscure since the author did not mention that how this relation is derived but only mentioned the term D and C are probably detectable. For better understandings, I would like to suggest the authors to make more clear clarifications.

Response: Thank you for the valuable suggestion. In the revised manuscript, we carefully take your suggestion into consideration through making clear clarifications and further performing calculations.

Firstly, we have made clearer clarifications on the origin of BPVE in edge-embedded structures. For edge-embedded structures, BPVE photocurrent density can still be expressed as $J_y^{\text{LBPVE}} = (\chi'_{\text{yxx}} + \chi'_{\text{yyy}})E_0^2 + (\chi'_{\text{yxx}} - \chi'_{\text{yyy}})E_0^2 \cos 2\alpha + 2\chi'_{\text{yxy}}E_0^2 \sin 2\alpha$. This is because the deduction of above equation does not require using any lattice symmetry. Hence, $D = (\chi'_{\text{yxx}} + \chi'_{\text{yyy}})E_0^2$ and $C = \sqrt{(\chi'_{\text{yxx}} - \chi'_{\text{yyy}})^2 + 4\chi'^2_{\text{yxy}}E_0^2}$. However, the second-order nonlinear DC tensor χ'_{ijk} of edge-embedded structures should be different from that of pure edges χ_{ijk} , since it depends not only on the type of materials but also on the geometry. For example, monolayer WS₂ flakes and WS₂ nanotubes are both non-centrosymmetric materials which theoretically support BPVE (Nature 570, 349-353, 2019). However, pronounced BPVE-induced photocurrents are only observed in WS₂ nanotubes. This suggests that the second-order nonlinear DC tensor of WS₂ nanotube should be significantly larger than that of monolayer WS₂ flake.

Hence, it is reasonable that the second-order nonlinear DC tensor χ'_{ijk} of edge-embedded structures is much larger than that of pure edges χ_{ijk} , since the geometry of engineered edge-embedded structures is different with that of pure edges.

Secondly, we have performed Finite-Difference Time-Domain (FDTD) simulations to compare the light absorption and power distribution in pure edges and edge-embedded structures (see **Fig. 4a,b**). Parameters of the model is based on the ReS₂/ReS₂ sample shown in Figure 2 in the main text. For 532 nm light, the absorption is 0.43 and 0.85 for pure edge and edge-embedded structure, respectively. Besides, the averaged power P_{average} in pure edge and edge-embedded region is similar. These simulation results prove that the small variation of light absorption and distribution in edge-embedded structures should not be able to significantly enhance the magnitude of BPVE-induced photocurrents.

Figure R4. a,b, Power distribution at pure edge (**a**) and edge-embedded structure (**b**).

In the revised manuscript, we have made clearer statement on the origin of BPVE in edge-embedded structures (see Page 9, Page 12 and Supplementary Note 1) and have included Fig. R4 in the Supplementary information (see Supplementary Fig. 13).

Second, the authors emphasize on the treatment of the edge-embedded region as a whole for the consideration of inversion symmetry breaking. The polarity of the BPVE photocurrent at the left and right edges are then fixed to have opposite directions. The authors aimed to provide a universal physical picture or mechanism for the photocurrent in most vdW edge-embedded structures. However, by comparing the results from ReS₂/ReSe₂ and WS₂/ReSe₂ samples, different signs for x_L and x_R can be found. As the charge transfer process at edges are attributed to enhance the photocurrent, it is necessary for the authors to pay more attention to the details of the photocurrent generation in different 2D materials-based edge-embedded structures since the charge transfer behavior in other 2D materials may be different.

Response: It is true that signs of photocurrents at x_L and x_R are different for ReS₂/ReS₂ and WS₂/ReS₂ structures. Based on your valuable suggestions, we further carefully compare photocurrent directions of two edges in all samples. We infer that the second-order nonlinear DC tensor χ'_{ijk} ($i/j/k$ represents $x, y,$ or z) plays an important role in determining the directions of BPVE-induced photocurrents I_{BPVE} . As discussed in the “Reply to Comment 1”, χ'_{ijk} depends on the geometry and orientations of edge-embedded structures. One example is for left and right edge-embedded structures in the same sample in the same xyz coordinate (see **Fig. R5**). Due to the reversed orientations, $\chi'_{yjk}(\text{left}) = -\chi'_{yjk}(\text{right})$, where j/k represents x or y . This results in opposite signs of photocurrents in left and right edge-embedded structures. We further compare two ReS₂/ReS₂ samples with different Re-chain directions. Their Re-chain directions are marked in **Fig. R6**. In sample 1, $I(x_L)$ and $I(x_R)$ are negative and positive, respectively. In sample 2, $I(x_L)$ and $I(x_R)$ are positive and negative, respectively. The different orientations of ReS₂ flakes in these two samples could probably result in different values and signs of χ'_{ijk} . For WS₂/ReS₂ edge-embedded structures, the configurations are different with ReS₂/ReS₂. Hence, it is possible that the sign and value of χ'_{ijk} in WS₂/ReS₂ edge-embedded structures is different from that in ReS₂/ReS₂ edge-embedded structures resulting in different photocurrent signs.

Figure R5. In the same sample and same xyz coordinate, left and right edges have reversed second-order nonlinear DC tensor.

Figure R6. a, Sample 1: ReS₂/ReS₂ sample with negative I_{ph} at x_L and positive I_{ph} at x_R . **b,** Sample 2: ReS₂/ReS₂ sample with positive I_{ph} at x_L and negative I_{ph} at x_R .

As pointed out by you that charge transfer processes may be different for different edge-embedded structures, we have calculated the band structure of bent WS₂ and further compared charge transfer processes between ReS₂/ReS₂ and WS₂/ReS₂ edge-embedded structures. As shown in **Fig. R7**, bent WS₂ and ReS₂ edge form type-I band alignment. Charge transfers can be enabled from WS₂ to ReS₂ edges, similar to that in ReS₂/ReS₂ edge-embedded structures. For other edge-embedded structures with different materials, charge transfer processes might be different and should be treated separately.

Figure R7. The first-principles calculations of density of states of ReS₂ edge, bent ReS₂ and bent WS₂.

In the revised manuscript, we have included above discussions in the main text (see Page 12 and 15). We have clearly stated that “It is worth noting that charge transfer processes are different for different edge-embedded structures. For other edge-embedded structures with different materials, charge transfer processes should be treated separately.” in the revised manuscript. Fig. R7 has been included in the Supplementary Fig. 12.

REVIEWERS' COMMENTS

Reviewer #1 (Remarks to the Author):

I have read the responses and the revised manuscript again. I find both them are satisfied and I recommend the publication of this version.

Reviewer #2 (Remarks to the Author):

In the revised version, the authors have reasonably addressed my previous comments. I recommend the publication of the manuscript.